# Androgen receptor (AR) antagonism triggers acute succinate-mediated adaptive responses to reactivate AR signaling

Neetu Saxena[1], Eliana Beraldi[1], Ladan Fazli[1], Syam Prakash Somasekharan[1] (ID), Hans Adomat[1], Fan Zhang[1], Chidi Molokwu[1], Anna Gleave[2], Lucia Nappi[2], Kimberly Nguyen[1] (ID), Pavn Brar[1], Nicholas Nikesitch[1], Yuzhuo Wang[1,2] (ID), Colin Collins[1,2], Poul H Sorensen[2] (ID) & Martin Gleave[1,2,*] (ID)

## Abstract

Treatment-induced adaptive pathways converge to support androgen receptor (AR) reactivation and emergence of castration-resistant prostate cancer (PCa) after AR pathway inhibition (ARPI). We set out to explore poorly defined acute adaptive responses that orchestrate shifts in energy metabolism after ARPI and identified rapid changes in succinate dehydrogenase (SDH), a TCA cycle enzyme with well-known tumor suppressor activity. We show that AR directly regulates transcription of its catalytic subunits (SDHA, SDHB) via androgen response elements (AREs). ARPI acutely suppresses SDH activity, leading to accumulation of the oncometabolite, succinate. Succinate triggers calcium ions release from intracellular stores, which in turn phospho-activates the AR-cochaperone, Hsp27 via p-CaMKK2/p-AMPK/p-p38 axis to enhance AR protein stabilization and activity. Activation of this pathway was seen in tissue microarray analysis on prostatectomy tissues and patient-derived xenografts. This adaptive response is blocked by co-targeting AR with Hsp27 under both *in vitro* and *in vivo* studies, sensitizing PCa cells to ARPI treatments.

**Keywords** Androgen receptor; Hsp27; prostate cancer; succinate dehydrogenase; succinate

**Subject Categories** Cancer; Metabolism

## Introduction

Despite frequent and durable responses of ARPI in advanced prostate cancer (PCa), progression to castration-resistant prostate cancer (CRPC) inevitably recurs, most often driven by AR reactivation involving genomic alterations, cross-talk signaling with kinase pathways, and coordinated action of different stress pathways (Wyatt & Gleave, 2015). ARPI-induced genomic modifications include AR amplification, promiscuous mutations, and constitutively active splice variants (e.g., AR-V7). In concert, stress chaperone proteins (e.g., Hsp27, clusterin) work together to stabilize AR protein complexes and promote nuclear translocation (Zoubeidi & Gleave, 2012), as well as mediating adaptive responses to ARPI that promote cell survival pathways and emergence of treatment resistance (Zoubeidi *et al*, 2010a; Zoubeidi & Gleave, 2012; Zhang *et al*, 2014; Wyatt & Gleave, 2015).

Metabolic remodeling is another stress response that functions to ensure sufficient energy and macromolecules for cancer cells under nutrient crisis. Prostate epithelial cells remodel their metabolism during state transition from benign to cancer to CRPC. Unlike citrate-secreting benign cells which are glycolytic, castration-sensitive PCa cells oxidize citrate in the TCA cycle to meet high energy demand of AR-driven, proliferating PCa cells (Hochachka *et al*, 2002; Singh *et al*, 2006). PCa cells are also highly addicted to lipid as an energy source. AR signaling orchestrates this metabolic remodeling by promoting mitochondrial biogenesis and lipogenesis via AMPK-PGC1α (Tennakoon *et al*, 2014) and regulating different glycolytic (HK2, PFK, GLUT1) (Massie *et al*, 2011) and lipid metabolism enzymes/ transcription factors (like FAS and SREBP) (Swinnen *et al*, 1997; Heemers *et al*, 2006). Androgen deprivation therapy (ADT) leads to mitochondrial dysfunction through altered mitochondrial mass as well as reduced expression and activities of various metabolic enzymes in the TCA cycle. These ADT-induced metabolic shifts provide a mechanistic link to the metabolic syndrome, which includes insulin resistance, cardiovascular risk, and obesity (Traish *et al*, 2011). A recent study (Schopf *et al*, 2020) demonstrated increased mitochondrial DNA load and rate of deleterious mutations causing mitochondria heteroplasmy associated with worse outcomes in PCa. As mitochondria are a major source of cellular electron transfer and ATP production, improved understanding of how ARPI rewires energy metabolism in PCa cells may define adaptive responses that support treatment resistance and may, in turn, be exploited therapeutically.

1   Vancouver Prostate Centre, Vancouver, BC, Canada
2   Department of Urologic Sciences, University of British Columbia, Vancouver, BC, Canada
    *Corresponding author. Tel: +1 604 675 8202; E-mail: m.gleave@ubc.ca

The TCA cycle is an integral part of aerobic respiration, generating ATP and precursors for building macromolecules and reducing agents that decrease oxidative stress. Succinate dehydrogenase (SDH) is the only TCA cycle enzyme connecting the TCA to the electron transport chain (ETC) as complex II (CII). SDH oxidizes succinate to fumarate in the TCA cycle and then further tunnels the released electrons to ubiquinone in the ETC (Hagerhall, 1997; Dalla Pozza *et al,* 2020). In humans, SDH is composed of two catalytic subunits, SDHA and SDHB, which reside in the mitochondrial matrix, and two membrane-spanning subunits, SDHC and SDHD (Fig 1A) and requires four assembly factors SDHAF1, SDHAF2, SDHAF3 and SDHAF4 for the assembly of whole complex (Sun *et al,* 2005; Moosavi *et al,* 2020). Genetic mutations in these subunits are associated with various cancers including paraganglioma, pheochromocytoma, gastrointestinal stromal tumors (GISTs), and renal cell and ovarian cancer (Bardella *et al,* 2011; Bezawork-Geleta *et al,* 2017; Moosavi *et al,* 2020). Reduced expression of these subunits, from genetic mutations or stressors that affect assembly and activity of SDH enzyme (Lemarie *et al,* 2011) (Chouchani *et al,* 2014; Mills & O'Neill, 2014), increase intracellular succinate levels (Bardella *et al,* 2011; Ariza *et al,* 2012). Increased succinate inhibits activity of the succinate precursor 2-oxoglutarate (2-OG)-dependent enzymes resulting in stabilization of HIF1α (Selak *et al,* 2005), increased DNA methylation (Yang & Pollard, 2013), and also post-translational modification of various proteins via succinylation (Zhang *et al,* 2011). Elevated succinate levels have also been reported to delay the initial differentiation of primed human pluripotent stem cells (TeSlaa *et al,* 2016). Recent study by Morris et al (Morris *et al,* 2019) suggests that tumor suppressor effects of p53 is mediated by 2-OG-dependent chromatin modification of 5-hydroxymethylcytosine (5hmC) resulting in transcriptional programing and therefore premalignant differentiation, which can be suppressed by increased succinate levels to promote malignant progression. In the present study, we set out to define early stress responses that orchestrate shifts in energy metabolism after acute, rather than chronic, ARPI, as rapid adaptation is key for survival under treatment stress and to support subsequent emergence of resistance. We identified acute decreases in SDH activity after ARPI with accumulation of succinate that triggered $Ca^{2+}$/p-CaMKK2/p-AMPK/p-p38-mediated phosphorylation of Hsp27 restoring cytoprotective AR activity. Co-targeted inhibition of Hsp27 with ARPI abrogated this metabolic adaptive response, uncovering potential therapeutic strategy.

# Results

## ARPI acutely represses SDH activity of CII by transcriptionally regulating its catalytic subunits via AR regulatory elements (AREs)

SDH enzyme, comprised of two catalytic subunits (SDHA, SDHB) and two membrane spanning subunits (SDHC, SDHD) (Fig 1A), plays a central role in energy homeostasis and has ascribed tumor suppressor (SDH) activity (Tretter *et al,* 2016; Bezawork-Geleta *et al,* 2017). To begin evaluation of ARPI on different metabolic pathways, we used tandem mass tagging (TMT) mass spec analysis of castrate-sensitive LNCaP cells treated with the AR antagonist, enzalutamide (ENZA), for 6, 12 and 24 h. Many metabolic pathways related to mitochondrial function and bioenergetics, including SDHA and SDHB subunits, were acutely altered by ARPI (Fig EV1A). Western blotting validated the above TMT results with reduced expression of SDH catalytic subunits SDHA and SDHB after ARPI using inhibitors against AR-ligand binding domain (ENZA, ODM-201), AR-DNA binding domain (VPC-14449) or androgen depletion (with CSS) in both LNCaP and LAPC4 cells (Fig 1B). AR silencing or knockdown (Fig EV1B) also reduced SDHA and SDHB protein levels similar to ARPI. Conversely, androgen (R1881) increased protein levels of both SDHA and SDHB (Fig 1B).

cBioPortal database analysis (Cerami *et al,* 2012; Gao *et al,* 2013) of different prostate adenocarcinoma studies indicates many alterations in SDHA and SDHB genes, particularly multiple shallow and deep deletions in *SDHB* gene (Fig EV1C,1D upper panel) as well as somatic missense and nonsense mutations in both *SDHA* and *SDHB* genes (Fig EV1D lower panel), suggesting that SDH activity can be altered in PCa. ARPI-induced decreases in SDH subunits expression in LNCaP and LAPC4 cells are, as expected (Lemarie *et al,* 2011), associated with reduced SDH activity (Fig 1C). Additionally, ARPI significantly increased intracellular succinate levels measured by LC-MS (Fig 1D) due to reduced SDH activity. Similar to protein levels, mRNA levels of catalytic subunits SDHA and SDHB were also reduced acutely after ARPI, suggesting that the AR might directly regulate their transcription (Figs 1E and EV1E). This notion was supported by the presence of AREs in *SDHA*, *SDHB*, *SDHD,* and *SDHAF2* genes promoters as derived from Genomatrix promoter analysis (Fig EV1F). Chipseq

**Figure 1. AR regulates SDH enzyme through AREs.**

A       Schematic representation of SDH composition (catalytic subunits: SDHA, SDHB, and membrane anchor subunits: SDHC, SDHD) and SDH activity.

B       ARPI (CSS, ENZA, ODM-201, 14449) decreased, while R1881 increased, SDHA and SDHB subunit protein levels in both LNCaP and LAPC4 cells. PSA is used as a positive control.

C, D    ARPI treatment (CSS or ENZA) for 12 h in LNCaP cells decreased SDH activity (C) and increased intracellular succinate levels (D) compared with vehicle treated control.

E       ENZA decreased (left panel), whereas R1881 increased (right panel), transcript levels of SDHA and SDHB in LNCaP cells.

F       Quantitative PCR on DNA immuno-precipitated with AR antibody in ChIP assay showed increased signal for SDHA, SDHB along with PSA after treatment with R1881 compared with androgen deprivation in LNCaP cells. PSA is used as positive control.

G       EMSA shows dose-dependent interaction between *SDHA* and *SDHB* AREs and AR-DNA binding protein, which was disrupted with mutated ARE probes (SDHA-M, SDHB-M). Quantification of binding of three independent experiments is shown at the bottom panel.

Data information: ENZA: enzalutamide, CSS: charcoal stripped serum, ODM-201: darolutamide. Data shown as mean ± SD of three independent experiments. Statistical analysis was performed using two tailed unpaired Student's *t*-test (G) or one-way ANOVA followed by Tukey's test (C, D, E, F). *$P < 0.05$, **$P < 0.01$, ****$P < 0.0001$ compared between groups. Exact *P* values are reported in the Appendix Table S7.

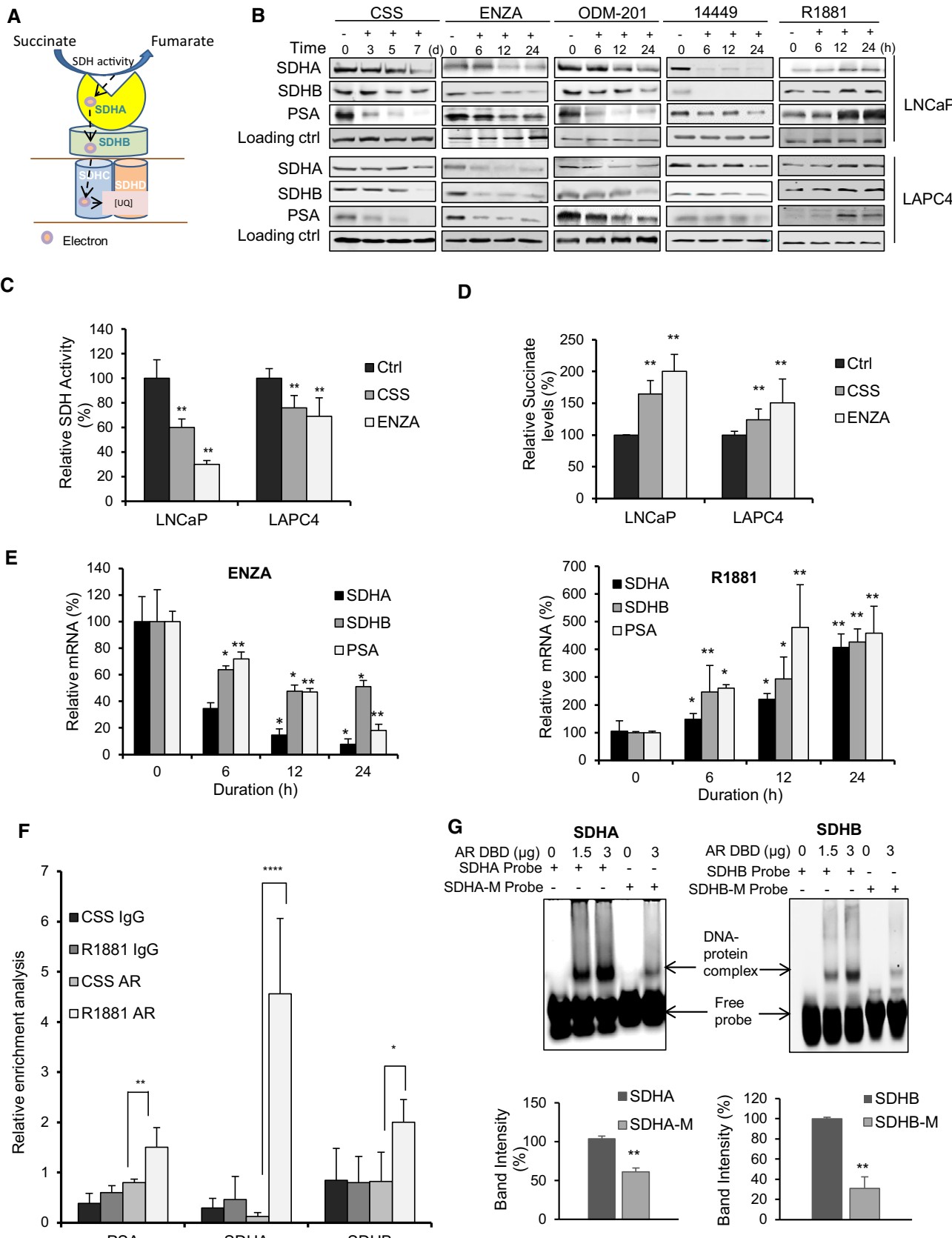

**Figure 1.**

analysis (Wilson *et al*, 2016) on LNCaP cells after R1881 treatment also predicted the presence of potential AREs in genes encoding the above subunits. We confirmed the presence of AREs in *SDHA* and *SDHB* promoters using chromatin immunoprecipitation (ChIP). ChIP analysis coupled with quantitative PCR confirmed interaction of AR protein with AREs in *SDHA* and *SDHB* promoters after R1881 (Fig 1F). Furthermore, binding of SDHA and SDHB AREs to AR DNA binding domain (DBD) was confirmed using electrophoretic mobility shift assays (EMSA) (Fig 1G). Both SDHA and SDHB ARE probes showed complex formation with AR DBD protein in a dose response manner. Conversely, site-directed mutations in the ARE dramatically decreased complex formation with AR-DBD. Collectively, these data indicate that *SDHA* and *SDHB* promoters have AREs and androgens/AR regulate their mRNA expression.

## Reduced SDH activity leads to increased AR expression and activity

Since SDH has tumor suppressor functions (Bezawork-Geleta *et al*, 2017), and AR is a major driver of PCa progression, we next evaluated whether ARPI-induced SDH repression contributes to re-activation of AR and castration resistance. SDHA or SDHB silencing increased, while their overexpression decreased, intracellular succinate levels in LNCaP cells (Fig EV2A and B). These data correlate loss- and gain- of SDHA/SDHB levels with SDH activity. In the presence of ENZA, SDH repression further increased succinate levels while SDH overexpression reduced succinate levels back toward baseline (Fig EV2C and D). SDH silencing dramatically increased AR protein levels in castrate-sensitive LNCaP and LAPC4, castrate-resistant C4-2, and ENZA-resistant MR49F PCa cell lines. Protein levels of the AR V7 splice variant and its activity in 22Rv1 cells were also increased after SDH inhibition (Figs 2A and EV2E). While ENZA acutely (up to 18 h) reduced AR protein levels, SDH repression in part rescued ENZA-mediated AR downregulation (Fig 2B), AR transcriptional activity (Fig 2C), and expression levels of AR-regulated genes in LNCaP cells (Fig 2D). ENZA-induced decreases in AR protein levels are consistent with prior reports (Gregory *et al*, 2001; Matsumoto *et al*, 2013) and are not accompanied by changes in AR mRNA levels after SDH repression (Fig 2D). AR protein levels and activity also increased when SDH activity was inhibited by the succinate analogue dimethylmalonate (DMM) (Fig EV2F and G). Conversely, overexpression of catalytic subunits SDHA and

SDHB reduced AR levels and activity in LNCaP (Fig 2E and F), as well as LAPC4 and 22Rv1, cells (Fig EV2H), supporting an SDH-specific effect. SDH-silenced cells retained higher survival after ENZA (Fig 2G), pointing to reduced SDH activity as a possible early response to restore AR activity and possibly survival after acute ARPI stress. In support of this, overexpression of these subunits abrogated this survival advantage under acute ENZA stress (Fig 2H). Overall, these data identify reduced SDH activity as a possible acute stress adaptive response to restore AR activity after ARPI. We next set out to define the mechanism underlying this adaptive response.

## Hsp27 supports AR upregulation post SDH repression

Because hypoxia enhanced AR protein upregulation after SDH repression (Fig EV3A), we initially speculated that HIF1α may be involved since it is stabilized by both hypoxia and SDH repression (Selak *et al*, 2005). HIF1α silencing, however, did not reverse SDH-repressed upregulation of AR transcript or protein levels (Fig EV3B). Because AR transcript levels do not change after SDH repression (Fig 2D), the translational inhibitor, cycloheximide, was used +/- SDHA/SDHB silencing to define possible post-translational effects. Compared with controls, AR protein stability increased by 20-30% after SDHA/SDHB silencing compared with ENZA alone, suggesting that SDH activity affects AR protein degradation rates (Fig 3A). SDH repression is known to increase p38 activity (Cervera *et al*, 2008; Chen *et al*, 2014), a kinase upstream of the stress chaperone Hsp27, which protects AR from MDM2-mediated proteasome degradation and promotes AR transcriptional activity (Xu *et al*, 2006; Zoubeidi *et al*, 2007). ENZA also increases p-p38 and p-Hsp27 levels as recently reported by Ware *et al* (preprint: Ware *et al*, 2020) (Fig EV3C). Interestingly, Hsp27 silencing reversed the increased levels of AR protein normally triggered by SDH repression (Fig 3B) and also inhibited the cell survival advantage mediated by SDH repression (Figs 3C and EV3D), leading to enhanced ENZA-induced apoptosis (Fig 3D). Inhibition of p38 kinase (which is upstream to Hsp27) by SB203580 or genistein also inhibited AR protein upregulation after SDH-repression (Fig 3E). Conversely, overexpression of CII catalytic subunit (SDHB) decreased p-p38, p-Hsp27, and AR levels (Fig EV3E). Collectively, these data suggest that ARPI-induced repression of SDH activity leads to adaptive increases in AR protein levels via increased p38 kinase phospho-activation of the AR co-chaperone, p-Hsp27. Unlike castration sensitive LNCaP cells,

**Figure 2. Inhibition of SDH activity increases AR protein levels and activity.**

A    SDHA and SDHB silencing significantly increased AR full length protein levels in LNCaP, LAPC4, C4-2, and MR49F cells as well as both AR full length and V7 protein levels in 22Rv1 cells.

B    ENZA acutely decreases AR levels (up to 18 h) while SDHB silencing rescued the early ENZA-mediated AR downregulation in LNCaP cells.

C    SDHA/SDHB silencing maintains higher levels of AR transcriptional activity during acute phase of ENZA stress (18 h) in LNCaP cells.

D    SDH repression during acute phase of ENZA stress increased transcription of different AR-regulated genes without altering AR mRNAs levels in LNCaP cells.

E, F    Overexpression of SDHA and SDHB subunits decreased AR protein levels (E) and AR transactivation (F) in LNCaP cells.

G, H    LNCaP cell viability is increased by SDHA/SDHB silencing (G) and reduced by SDHA/SDHB overexpression (H) compared with siScr and empty vector, respectively.

Data information: ENZA: enzalutamide. Data shown as mean ± SD of three independent experiments. Statistical analysis was performed using two tailed unpaired Student's *t*-test (D) or one-way ANOVA followed by Tukey's test (C, F, G, H). *$P < 0.05$ and **$P < 0.01$ compared between groups. Exact *P* values are reported in the Appendix Table S7.

Source data are available online for this figure.

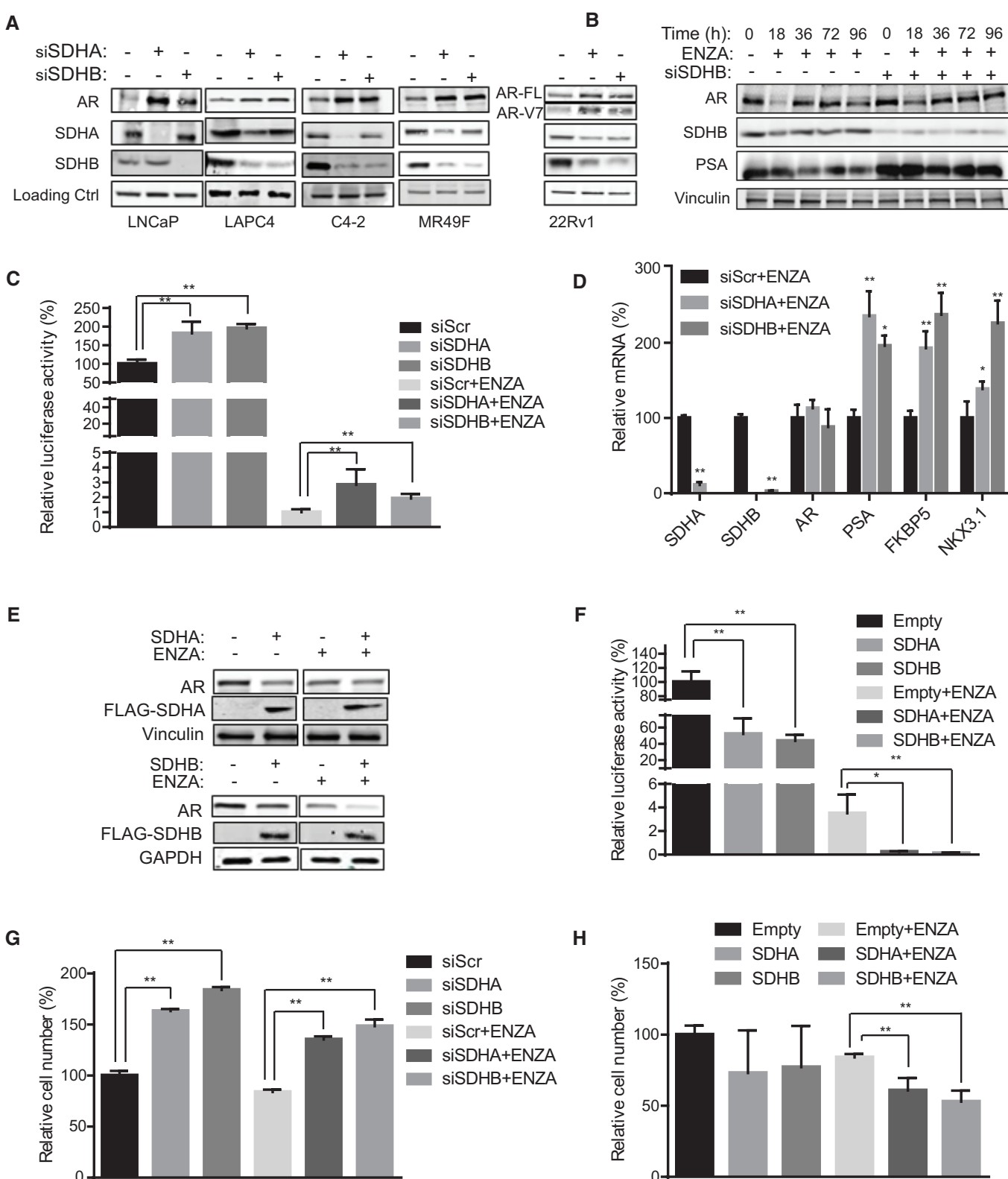

Figure 2.

which lose AR activity in response to androgen deprivation (CSS treatment), castration-resistant V16D and ENZA-resistant MR49F cell lines, which maintain AR activity independent of androgen availability, demonstrate more SDH activity suggesting that reactivation of AR activity in CRPCs may help restore SDH activity (Fig EV3F).

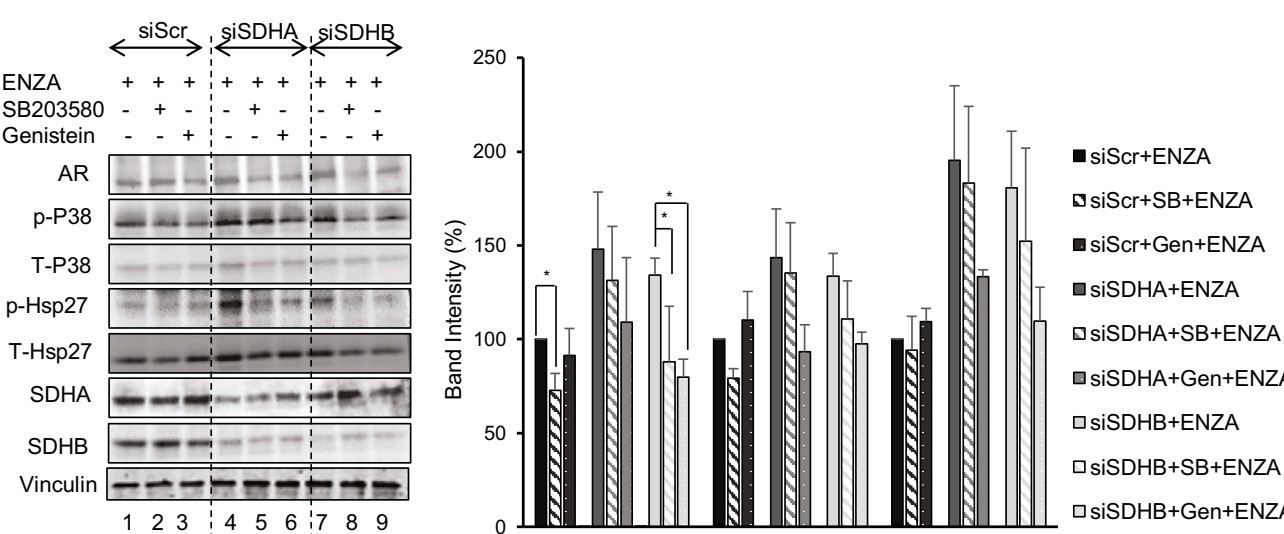

**Figure 3.**

**Figure 3.   Hsp27 regulates the expression and activity of AR post SDH repression.**

A   LNCaP cells silenced for SDHA or SDHB showed approximately 20-30% longer AR half-life compared to ENZA alone in the presence of 50 µg/ml cycloheximide (left panel). Right panel shows quantification of AR protein normalized by loading control.
B   Co-silencing of SDHB and Hsp27 reversed siSDHB-mediated AR protein upregulation in LNCaP cells.
C   SDH-repressed LNCaP cells were able to maintain higher proliferation rate particularly during ENZA stress with or without Hsp27 silencing in real-time cell proliferation assay.
D   Hsp27 silencing in LNCaP cells showed dramatically high levels of cleaved-PARP compared with ENZA, which were rescued to some extent by SDH-repression.
E   Treatment with p-p38 inhibitors: SB203580 (10 µM) or Genistein (20 µM) for 24 h reduced AR upregulation induced by SDH repression in LNCaP cells (left panel). Bar graph in the right panel shows band intensity normalized to loading control.

Data information: ENZA: enzalutamide. Data shown as mean ± SD of three independent experiments. Statistical analysis was performed using one-way ANOVA following Tukey's test for panels C (end points) and E . $*P < 0.05$, $**P < 0.01$ and $***P < 0.001$ compared between groups, Exact P values are reported in the Appendix Table S7. Source data are available online for this figure.

## Accumulated succinate triggers calcium ion flux to activate p-CaMKK2/p-AMPK/p-p38/p-Hsp27 axis to increase AR

Since AR is known to fuel different bioenergetics pathways (Swinnen *et al*, 1997; Heemers *et al*, 2006; Massie *et al*, 2011; Tennakoon *et al*, 2014), and SDH is a key TCA cycle enzyme, we investigated effects of ARPI-mediated SDH repression on energy metabolism. Extracellular flux analysis using a glycolysis stress test measures cellular adaptive capacity to switch from oxidative phosphorylation to glycolysis under stress conditions. SDH-repressed cells demonstrated higher bioenergetic profiles by primarily increasing glycolysis (up to 100% increase) and to a lesser extent, mitochondrial respiration (~50%), compared with ENZA alone (Fig 4A and B). ENZA-induced changes in glycolytic capacity, defined as the maximum extracellular acidification rate (ECAR) achieved by cells after inhibition of oxidative phosphorylation by ATP synthase inhibitor oligomycin, was doubled in SDH repressed cells suggesting that SDH repression helps shift the dependency on ATP production from mitochondrial respiration to glycolysis. LC-MS analysis on cellular extracts also identified increased levels of lactic acid production after SDH repression, consistent with increased glycolysis; in contrast, overexpression of CII catalytic subunits (SDHA, SDHB) decreased lactic acid levels (Fig EV4A), suggesting that SDH repression post ARPI helps PCa cells adapt to energy stress by promoting glycolytic capacity. In agreement with this, SDH-repressed cells produced more ATPs after ENZA compared with controls (Fig EV4B).

AMP-activated protein kinase (AMPK) is phosphorylated in response to increased AMP/ATP ratios to re-establish energy homeostasis by increasing catabolic while inhibiting anabolic pathways, and supporting cell survival via p-p38 (Xi *et al*, 2001; Chaube *et al*, 2015). SDHA and SDHB silencing increased AMPK phosphorylation and enhanced levels of p-p38 and p-Hsp27; AMPK silencing or inhibition (using Dorsomorphin similar to p38 kinase inhibition in Fig 3E) reduced p-p38/p-Hsp27 levels and diminished upregulation of AR (Figs 4C and EV4C) and cell survival (Fig EV4D) after SDH repression under ENZA stress. AMPK is therefore a key upstream sensor linking this adaptive response to ARPI-induced energetic stress and restoration of AR activity. LKB1 and CaMKK2 are two major kinases regulating AMPK phosphorylation and activity (Hawley *et al*, 2005; Marcelo *et al*, 2016). Androgens are known to promote CaMKK2 activity and thereby fuel energy metabolism through p-AMPK (Massie *et al*, 2011; Karacosta *et al*, 2012). Therefore, we next characterized effects of ARPI-induced changes in SDH on CaMKK2 in LNCaP cells. SDHA and SDHB subunit overexpression decreased, while their silencing increased, p-CaMKK2 levels (Fig 4D and E). Dimethyl succinate (DMS), which mimics succinate accumulation after SDH repression, also increased p-CaMKK2/p-AMPK/p-p38/p-Hsp27 axis (Fig EV4E). The CaMKK2 inhibitor STO-609 (Tokumitsu *et al*, 2002) dramatically reduced CaMKK2 and AMPK phosphorylation and downstream signaling cascade after DMS or ENZA (Fig EV4E and F). Moreover, CaMKK2 silencing (Fig 4E) or treatment with STO-609 (Fig 4F) similarly inhibited AR upregulation after SDHA or SDHB silencing. Collectively, these data confirm that AMPK phosphorylation triggered by succinate accumulation post ARPI is mediated by CaMKK2.

As CaMKK2 is a calmodulin-dependent kinase activated by calcium ions (Hawley *et al*, 2005), we next evaluated if succinate

**Figure 4.   Succinate increases intracellular calcium flux to activate p-CaMKK2/p-AMPK in LNCaP cells.**

A   Extracellular flux analysis by measuring oxygen consumption rate (OCR) (left panel) and extracellular acidification rate (ECAR) (right panel) in LNCaP cells indicates that SDHA/SDHB silencing increased both activities compared with ENZA alone.
B   Comparison of OCR vs ECAR clearly shows significant upregulation of glycolytic capacity (2 fold) in SDH-repressed cells compared with ENZA alone.
C   Silencing of SDHA/SDHB subunits in LNCaP cells increased p-AMPK, as well as p-p38/p-Hsp27 levels. Co-silencing of AMPK inhibited the upregulation of AR mediated by SDH repression.
D   *SDHA* and *SDHB* overexpression decreased p-CaMKK2 and p-AMPK levels in LNCaP cells.
E   *SDHA* and *SDHB* silencing increased AR, p-CaMKK2 and p-AMPK axis in LNCaP cells, which was inhibited by CaMKK2 silencing.
F   Treatment with 5 µM STO-609 for 24 h in LNCaP cells inhibited AR upregulation induced by SDHA/SDHB subunit silencing.
G   Thapsigargin, DMM, DMS, and ENZA treatment for the indicated time points increased intracellular calcium flux in LNCaP cells as visualized with calcium sensitive Fluo4-AM probe under confocal microscope.

Data information: ENZA: enzalutamide, 2-DG: 2-deoxyglucose, DMM: Dimethyl malonate, DMS: Dimethyl succinate. Data shown as mean ± SEM of three independent experiments. Scale bar: 10 µm.
Source data are available online for this figure.

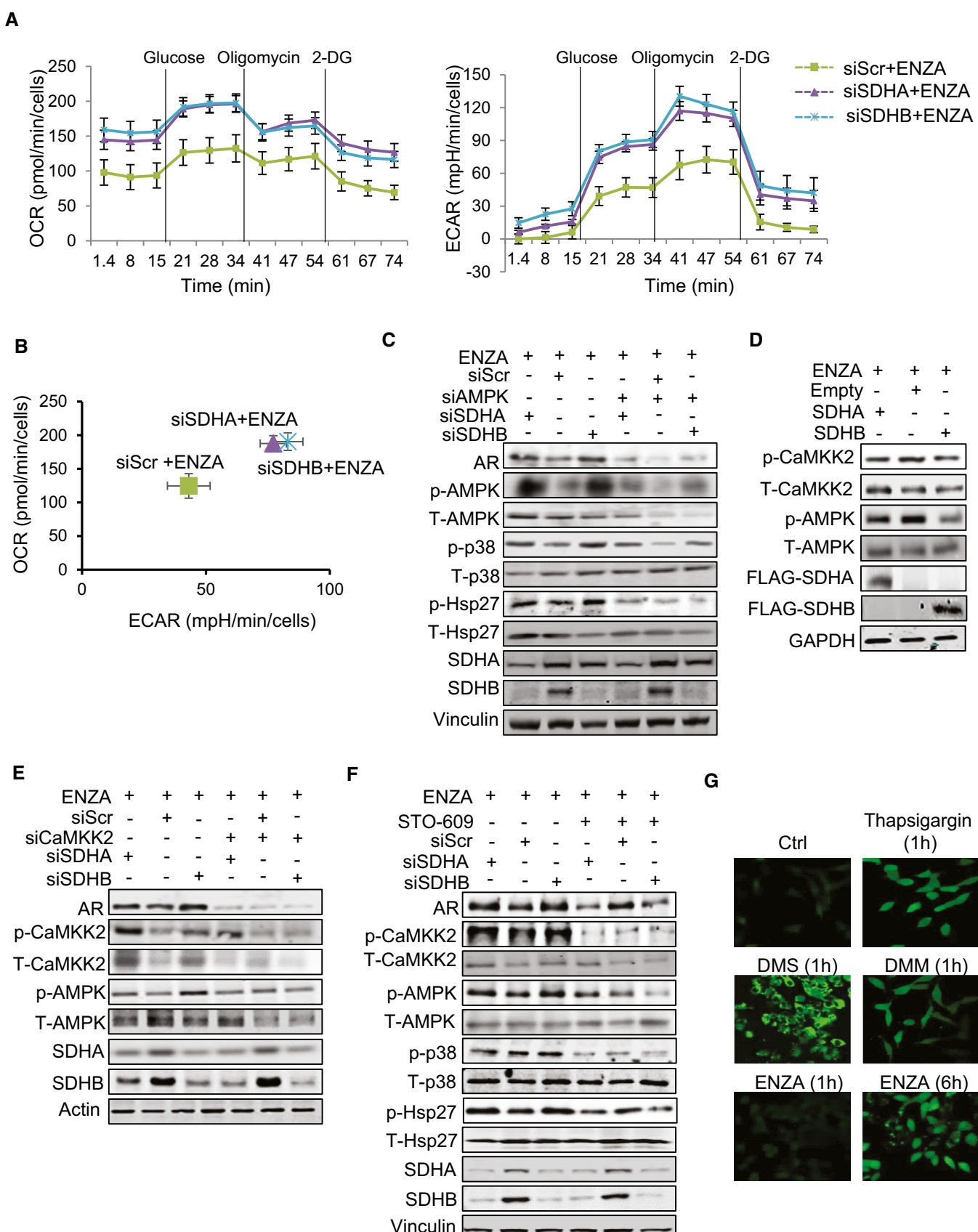

**Figure 4.**

promotes intracellular calcium ion flux. Treatment with both DMS (cell permeable succinate), DMM (SDH activity inhibitor), and ENZA all significantly induced intracellular calcium ion flux in the presence of extracellular $Ca^{2+}$ chelator EDTA similar to thapsigargin (Fig 4G), an inhibitor of ATP-dependent $Ca^{2+}$ SERCA pumps located at the ER (Lytton et al, 1991). Intracellular calcium flux increased rapidly with DMM and DMS (1 h) compared to ENZA (6h) (Fig 4G), where succinate accumulation is a secondary effect of SDH activity inhibition post ENZA. Overall, these data suggest that succinate accumulation post ARPI acts as a secondary messenger to increase calcium ion flux to activate p-CaMKK2/p-AMPK signaling.

### In vivo study in human and xenografts support AR-SDH loop through p-CaMKK2/p-AMPK/p-p38/p-Hsp27 axis

Immunohistochemistry (IHC) staining of tissue microarrays (TMA) spotted with clinical PCa tissues demonstrated that SDHA and AR expression was significantly reduced in patients treated with neoadjuvant ADT (6 months or less), compared with those who had prostatectomy alone (Figs 5A–C and EV5A). Consistent with cell line data, reduced SDH levels in neoadjuvant-treated prostatectomy tissue was associated with higher levels of p-CaMKK2/p-AMPK/p-p38/p-Hsp27. AR expression and activity increased again in CRPC samples (Figs 5A–C and EV5A), which then serves to reactivate CaMKK2 and subsequent downstream axis (Massie et al, 2011; Karacosta et al, 2012). In contrast, most neuroendocrine PCa (NEPC) samples with lower AR levels had reduced p-Hsp27 levels (Fig EV5B). Similarly, NEPC PDX models (Lin et al, 2014) with undetectable AR levels showed significantly reduced p-Hsp27 levels (Fig EV5C and D), which could be a limiting factor for AR stabilization despite having reduced SDHA and elevated or similar p-CaMKK2/p-AMPK/p-p38 axis compared with castrate-sensitive or castrate-resistant PCa. GSEA analysis on AR-positive CRPC vs AR-negative NEPC demonstrated enrichment of signature pathways for androgen response in CRPC and G2/M checkpoint and E2F signaling in NEPC (Fig EV5E). Additionally, SDHB-related pathways (oxidative phosphorylation, adipogenesis, fatty acid metabolism) were enriched in CRPC but not NEPC (Fig EV5F). Conversely, NEPC showed enrichment for hypoxia and glycolysis pathways (Fig EV5E) which are known to be upregulated with SDH repression (Cervera et al, 2008; Saxena et al, 2016). Collectively, these studies provide clinical relevance that supports experimental cell line studies. An early succinate-mediated response to AR antagonism, initiated by reduced SDH activity, triggers calcium flux to activate the p-CaMKK2/p-AMPK/p-p38 axis, leading to phosphoactivation of Hsp27 that functions to stabilize and increase AR protein levels and activity. Restored AR activity again increases SDH levels and p-

CaMKK2 levels leading to phosphorylation of AMPK and downstream p-p38/p-Hsp27 axis to refuel the energy production and AR signaling. This finding was supported by in vivo study with LNCaP xenografts, which demonstrated reduced levels of SDH protein and activity along with AR under acute stress of castration; SDH levels and activity were restored in tumors demonstrating AR reactivation with castration resistance (Fig 6A). In our recent study, we reported ivermectin (IVM) as a small molecule inhibitor which binds to the phosphorylation site of 24 monomer complex of Hsp27 and blocks its phosphorylation mediated by MAPKAP2 (Nappi et al, 2020). Inhibition of Hsp27 phosphorylation by IVM decreased AR activity (Fig 6B) and hence cell proliferation (Fig 6C) in LNCaP cells when combined with ENZA. Similarly, IVM also inhibited SDH-repression mediated cell survival benefit in LNCaP cells (Fig 6D). Additionally, unlike vehicle-treated castrated samples, IVM-treated samples demonstrated reduced PSA levels (Fig 6E upper panel), tumor growth (Fig 6E lower panel), p-CAMKK2/p-p38/p-Hsp27 axis activity, and AR levels compared with pre-castration samples (Fig 6F).

Collectively, these data define an acute adaptive response whereby ARPI-induced decrease in SDH activity triggers rapid succinate accumulation and p-AMPK activation via p-CaMKK2, which leads to p-38 kinase-mediated Hsp27 phosphorylation and stabilization of AR protein levels and activity. Combining ARPI with Hsp27 inhibition offers potential strategies to block this adaptive response and enhance the activity of ARPI through downstream effect on AR stabilization (Fig 7).

## Discussion

AR signaling drives energy metabolism to promote cell secretory, survival, and proliferation pathways that support PCa progression. While ARPI disrupt these pathways, adaptive stress-responses are induced that reactivate AR-regulated cell survival and therapy resistance. Many varied mechanisms, including changes in mRNA translation (Leprivier et al, 2015), selective upregulation of stress adaptor proteins (Zoubeidi & Gleave, 2012), and activation of alternative signaling pathways (Zoubeidi et al, 2010a; Carver et al, 2011; Zhang et al, 2014) support stress adaptation in cancer, but acute changes in bioenergetic pathways in AR-driven PCa after ARPI remain poorly defined. As energy is an immediate requirement for cell survival under stress conditions, we focused on acute and early changes in energy metabolism after ARPI stress. This study defines, for the first time, novel cross-talk between AR and a mitochondrial SDH enzyme whereby ARPI inhibition of SDH activity triggers activation of the AR co-chaperone, Hsp27 through $Ca^{2+}$/p-CaMKK2/p-AMPK/p-p38 axis to restore AR activity. Interestingly targeted inhibition of the

**Figure 5. IHC of human prostate TMA identifies increases in p-CAMKK2/p-AMPK/p-p38/p-Hsp27 axis after ADT.**

A–C  Human patient samples demonstrate relative differences in the immunoexpression of AR, SDHA, p-CaMKK2/p-AMPK, p-p38, and p-Hsp27 in untreated (n = 70), neoadjuvant-treated (NHT) (n = 130) and CRPC (n = 24) samples. Representative images (A), dot plots (B), compiled bar graph (C) support the AR-SDH loop where inhibition of AR axis in NHT samples reduced SDHA levels along with upregulation of p-CaMKK2/p-AMPK/p-p38/p-Hsp27 axis. AR-positive CRPC samples restored SDHA as well as the whole axis.

Data information: Data shown as mean ± SEM. Statistical analysis was performed using one-way ANOVA followed by Tukey's test. **$P$ < 0.01; ***$P$ < 0.001 and ****$P$ < 0.0001. Scale bar: 100 μm. All the TMAs were constructed from minimum 2 cores per patient.
Source data are available online for this figure.

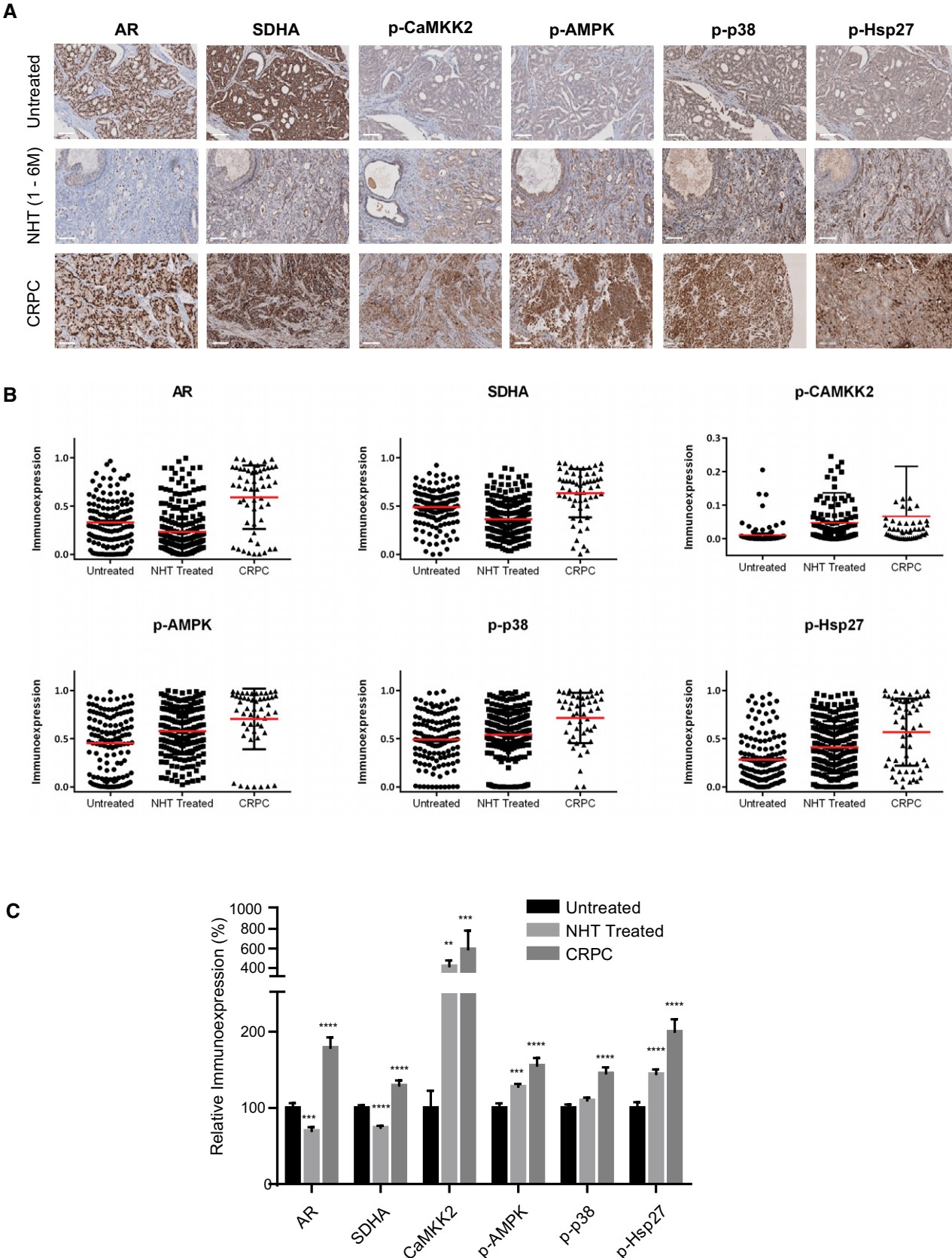

**Figure 5.**

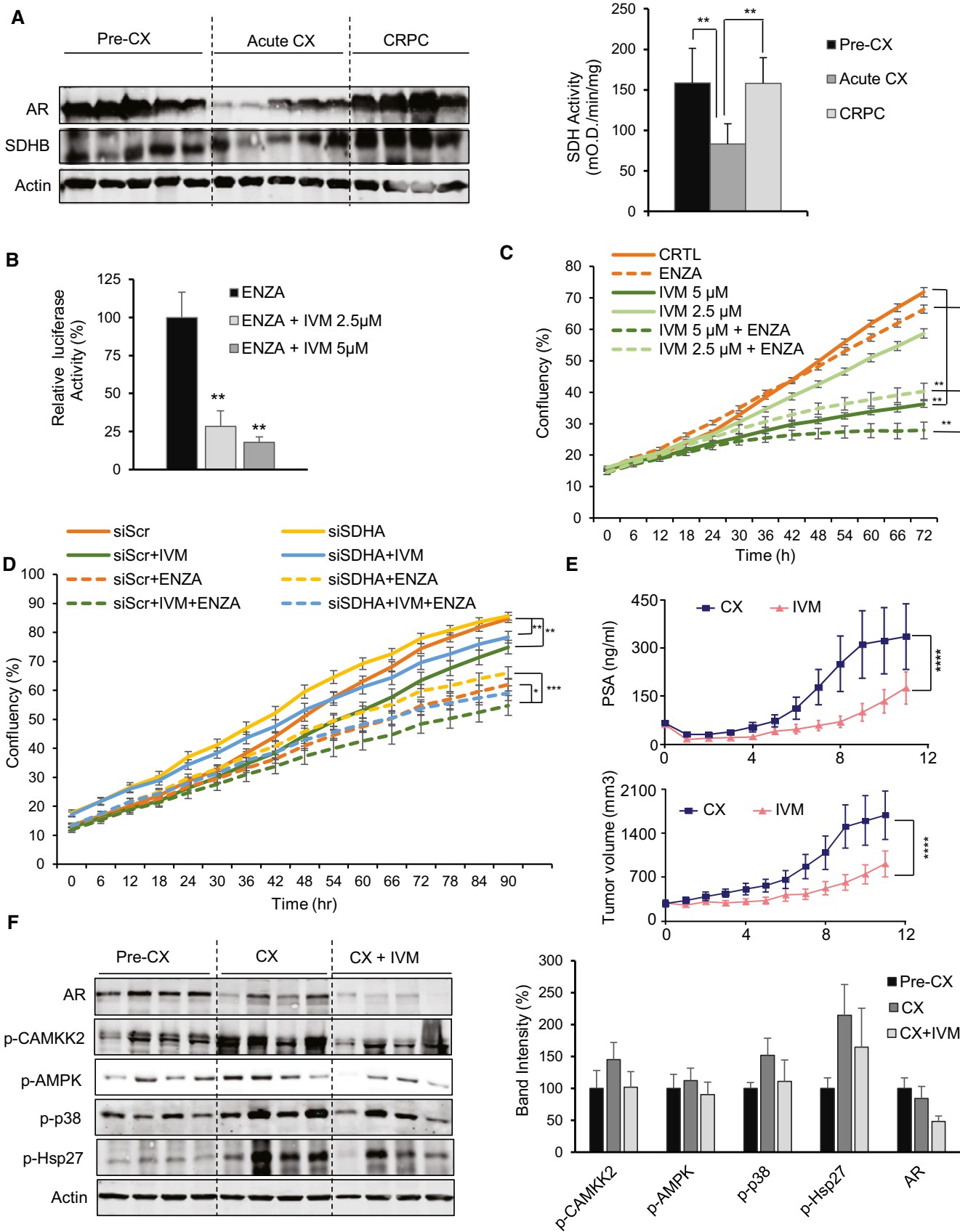

**Figure 6.**

**Figure 6. IVM decreases AR levels and activity in both *in vitro* and *in vivo* studies.**

A   In LNCaP xenografts, AR and SDHB protein levels (left panel) and SDH activity (right panel) were reduced acutely after castration (3 days) but returned to baseline after castration resistance (3 months). SDH assay was performed on tumor tissues from 5 mice per group.

B   IVM reduced AR activity in combination with ENZA in LNCaP cells.

C   ENZA reduced cell proliferation more potently when combined with IVM in LNCaP cells.

D   IVM reversed the cell proliferation benefit mediated by SDH repression in LNCaP cells.

E   LNCaP xenografts treated with IVM ($n = 10$) (10mg/kg) demonstrated lower serum PSA values (upper panel) and growth of tumors after castration (lower panel) compared with vehicle ($n = 10$). PSA was measured every week. The data have been taken from our published study in Nappi *et al* (2020).

F   IVM treatment reduced p-Hsp27 levels and therefore AR protein level, with subsequent p-CAMKK2 axis activation in castrated LNCaP xenografts. Bar graph in the right panel shows quantification of data from Western blot normalized to loading control.

Data information: ENZA: enzalutamide, IVM: ivermectin. Data shown as means $\pm$ SEM (C, D, E) or means $\pm$ SD (A, B, F). *In vitro* data represent three independent experiments. Statistical analysis was performed for the last data point (C, D). *$P < 0.05$, **$P < 0.01$ and ***$P < 0.001$ compared between groups, one-way ANOVA followed by Tukey's test. Exact *P* values are reported in the Appendix Table S7.

Source data are available online for this figure.

downstream effector Hsp27 blocks this feed-forward loop that restores AR signaling and could therefore be exploited to enhance anti-cancer activity of ARPI.

Benign prostate epithelial cells depend on malate, glutamate, and other amino acids for ATP production through complex I in ETC by channeling pyruvate for citrate production and secretion. A recent study by (Schopf *et al*, 2020) found that deleterious mutations in complex I in PCa inhibits oxidation of these substrates and instead forces oxidation of succinate (and to some extent pyruvate) through complex II for ATP production. The substrate specificity shifts with increasing tumor grade from glutamate, malate to succinate, pyruvate, suggesting PCa cells are more dependent on SDH activity for energetics to compensate the loss of complex I activity and are therefore more susceptible to succinate accumulation and signaling after ARPI. AR transcriptionally upregulates various enzymes in glycolysis and the TCA cycle either directly or via the AMPK/PGC1α axis (Tennakoon *et al*, 2014). SDH enzyme is situated at the intersection of TCA cycle and the ETC, acting as an oxygen sensor to promote shifts toward glycolysis from mitochondrial respiration during hypoxia or energetic stress. We identify SDH as a direct downstream target of AR with AR-response elements in the promoter region of 3 subunits (SDHA, SDHB, SDHD), one assembly factor (SDHAF1), and a chaperone (HSCB) required for delivery of [Fe-S] cluster in SDHB subunit (Saxena *et al*, 2016; Wilson *et al*, 2016). SDH is the only respiratory complex of ETC which is encoded by a nuclear, rather than mitochondrial, genome (Yankovskaya *et al*, 2003; Rutter *et al*, 2010). Direct transcriptional regulation of SDH enzyme by AR may impart privileged control of energetics necessary for AR-driven tumor progression. We validated AREs in *SDHA* and *SDHB* promoters and demonstrate reduced expression of these subunits post ARPI.

cBioPortal analysis demonstrated significant proportion of somatic mutations and deletions, particularly in *SDHB*, in various PCa studies. These genomic alterations highlight a tumor suppressive activity of SDH subunits, where loss of SDH activity increases intracellular succinate, an oncometabolite with key roles in HIF1α stabilization, post-translational modifications, and genome-wide methylation (Tretter *et al*, 2016; Bezawork-Geleta *et al*, 2017). Various signaling pathways like p38 and EMT are also activated in these cancers (Cervera *et al*, 2008; Chen *et al*, 2014). High succinate levels are associated with tissue injury or inflammation, where succinate functions as a secondary messenger (Chouchani *et al*, 2014; Mills & O'Neill, 2014; Moosavi *et al*, 2020). Reduced expression of SDH

subunits post ARPI may hamper assembly of the SDH enzyme complex (Lemarie *et al*, 2011), leading to reduced SDH activity and increased succinate levels post ARPI. We found that SDH repression inhibits proteasomal degradation of AR by upregulating p-Hsp27 (Zoubeidi *et al*, 2007), findings consistent with studies in breast cancer, where SDH repression can also upregulate estrogen receptor (ER) β expression (Manente *et al*, 2013) and promote mitochondrial biogenesis via PGC1α (Chen *et al*, 2009).

Stress chaperones like Hsp27, clusterin, and YB1 are induced by treatment to facilitate cell survival pathways activation (Zhang *et al*, 2014; Somasekharan *et al*, 2015; Wyatt & Gleave, 2015). For example, Hsp27 regulates many hallmarks of cancer and is associated with poor outcome and treatment resistance. We previously reported roles for Hsp27 in proteostatic (Zoubeidi & Gleave, 2012), IL-6 (Shiota *et al*, 2013), and IGF-1 (Zoubeidi *et al*, 2010b) survival signaling, as well as specific oncoprotein (e.g., BRAF, MET, AR, EGFR) interactions (Zoubeidi *et al*, 2007; Konda *et al*, 2017). SDH repression has been reported to increase p38 phosphorylation (Cervera *et al*, 2008; Chen *et al*, 2014), a major kinase that phosphorylates Hsp27 under stress conditions. Increased Hsp27 phosphorylation after SDH repression could therefore represent another adaptive mechanism during energy stress. Our study delineates, for the first time, a feed-forward loop whereby ARPI-induced SDH repression leads to succinate-mediated calcium flux, which activates CaMKK2-mediated phosphorylation of the energy sensor p-AMPK to, in turn, trigger p38 phosphorylation of the AR co-chaperone, Hsp27. CaMKK2 activity is induced by AR agonists and suppressed by ADT to regulate AMPK phosphorylation and thereby energy production downstream of AR (Massie *et al*, 2011; Karacosta *et al*, 2012). However, SDH repression post ARPI appears to reactivate CaMKK2 and therefore p-AMPK (and subsequently p-38/p-Hsp27) via accumulation of succinate. Unlike androgens, which induce intracellular calcium via G-protein coupled receptors (Sun *et al*, 2006), ENZA and succinate increase calcium ion release from internal stores in a manner similar to thapsigargin, inhibiting ATP-dependent $Ca^{2+}$ SERCA pumps in the endoplasmic reticulum (ER) to release calcium ions into cytosol. Cross-talk between mitochondria and ER is quite evident in $Ca^{2+}$ homeostasis, autophagy, apoptosis, energy, and phospholipid metabolism, mitochondrial biogenesis, ER stress and unfolded protein responses (Rowland & Voeltz, 2012; Li *et al*, 2013; Rainbolt *et al*, 2014; Gomez-Suaga *et al*, 2017). Mitochondrial accumulation of succinate may act as a second messenger to trigger release of calcium ions from ER and rapidly activate cell

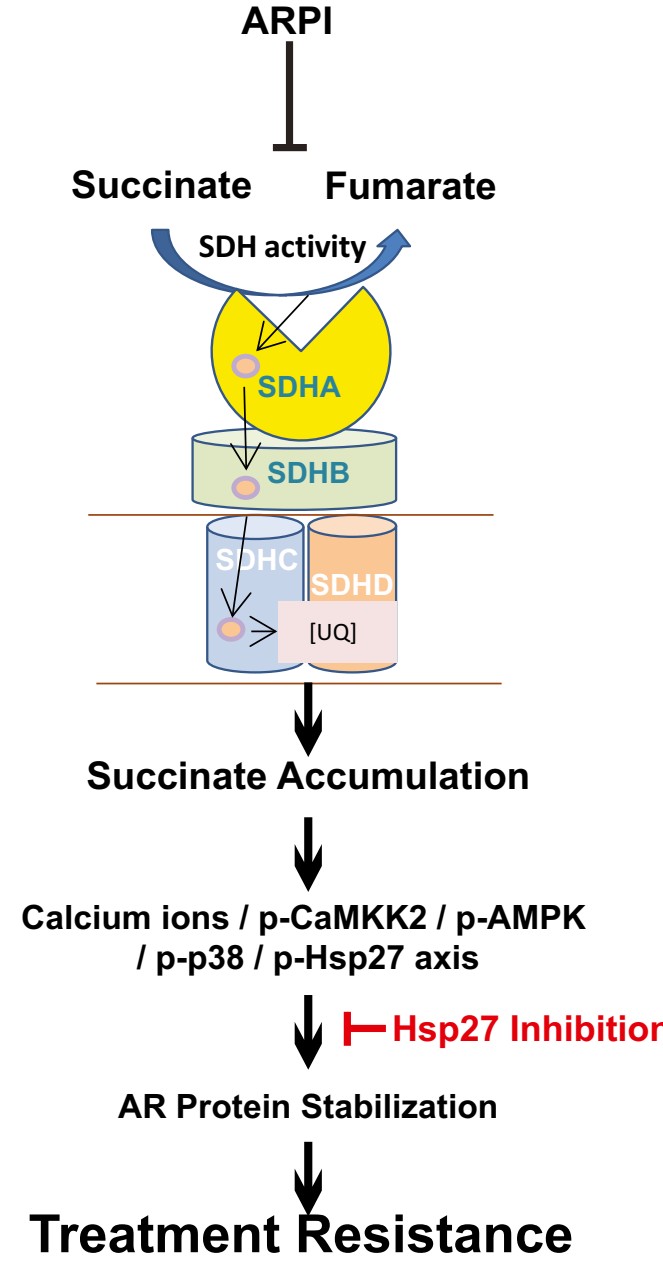

**ARPI**

**Succinate    Fumarate**

**SDH activity**

**SDHA**

**SDHB**

**SDHC**

**SDHD**

[UQ]

**Succinate Accumulation**

**Calcium ions / p-CaMKK2 / p-AMPK / p-p38 / p-Hsp27 axis**

**Hsp27 Inhibition**

**AR Protein Stabilization**

**Treatment Resistance**

**Figure 7.  AR-SDH cross-talk mediates adaptive response to ARPI through p-HSP27.**

ARPI acutely inhibits SDH activity, triggering the $Ca^{2+}$/p-CaMKK2/p-AMPK/p-p38/p-Hsp27 axis through accumulation of the oncometabolite succinate, which supports AR protein stabilization and activity. This AR-SDH feed-forward loop can be blocked by inhibiting Hsp27 activity.

signaling networks. Collectively, our study suggests that SDH repression post ARPI induces a $Ca^{2+}$/p-CaMKK2/p-AMPK/p-p38/p-Hsp27 pathway which restores AR activity to facilitate treatment resistance. Once restored, AR can again increase expression and SDH activity to re-fuel energy metabolism as supported by increased SDH levels and activity in CRPC after long-term castration which had restored AR levels.

IHC studies of clinical samples also demonstrate reduced SDHA levels with activation of p-CaMKK2/p-AMPK/p-p38/p-Hsp27 axis after neoadjuvant ADT. Restoration of AR levels in CRPC samples refuels energy metabolism by increasing SDH levels and p-CaMKK2-mediated axis (Massie *et al*, 2011; Karacosta *et al*, 2012). Interestingly, in contrast to castrate-resistant adeno-PC, the AR pathway is quiescent in highly glycolytic neuroendocrine PCa (NEPC) (Lin *et al*, 2014; Choi *et al.*, 2018), characterized by low levels of SDH that could lead to stabilization of HIF1α to support glycolysis, angiogenesis and invasion. Low levels of p-Hsp27 in NEPC, in contrast to adeno-CRPC, further support its role in SDH-repression mediated AR restoration. We previously reported anti-cancer activity of Hsp27 inhibition in a randomized clinical trial phase II where PSA response rates were doubled in CRPC patients co-treated with apatorsen (OGX-427, a second-generation antisense oligonucleotide targeting Hsp27 mRNA) than prednisone alone (Yu *et al*, 2018). Recently we identified ivermectin as a small molecule inhibitor of Hsp27 phospho-activation that blocks downstream survival signaling and potentiates ARPI in PCa cells. Ivermectin significantly reduced AR FL or V7 levels in LNCaP and 22Rv1 cells and xenografts to reduce tumor progression in combination with ARPI (Nappi *et al,* 2020). The present study further defines the therapeutic effect observed in our earlier studies. IVM blocks early SDH-mediated AR stabilization post ARPI through inhibiting Hsp27 phosphorylation and hence disrupts the AR-SDH feed-forward loop.

In summary, this study defines novel cross-talk between AR, energy metabolism, and adaptive responses to ARPI through SDH enzyme. AR directly regulates transcription of catalytic subunits of SDH enzyme via AREs; ARPI-reduced SDH activity triggers succinate accumulation and release of intracellular calcium ion leading to CaMKK2 activation of p-AMPK/p-p38/p-Hsp27 that function to stabilize and restore AR protein levels and activity. These mechanisms provide insights into metabolic stress adaptation and highlight how co-targeting the AR with Hsp27 blocks this adaptive response to enhance ARPI activity in PCa.

## Materials and Methods

### Human subjects

Human specimen obtained from Vancouver Prostate Centre (VPC) tissue bank were used to develop TMA to study the progression and trans-differentiation of the tumors based on their immunoreactivity against relevant markers. TMA was constructed from 240 radical prostatectomies (RPs) and transurethral resection of the prostate (TURPs) specimens. Master clinical data of VPC tissue bank were used to select the specimens. No patient was specifically recruited for TMA. The informed consent was obtained from all the subjects, and the experiments conformed to the principles set out in the WMA Declaration of Helsinki and the Department of Health and Human Services Belmont Report. The study protocol was approved by University of British Columbia, Office of Research Ethics, Clinical Research Ethics Board, UBC CREB NUMBER: H09-01628.

### Cancer cell lines

Castration-sensitive human prostate cancer LNCaP cells and castration-resistant C4-2 and 22Rv1 cells were kindly gifted by Dr. Leland

W.K. Chung (Hsieh *et al*, 1993). Castration-resistant V16D and ENZA-resistant MR49F cell lines were generated as a result of serial passage of LNCaP xenografts (Kuruma *et al*, 2013). LN95 and LAPC4 cells were kindly provided by Dr. Stephen Plymate and Dr. Charles Sawyer, MSKCC, respectively. LNCaP, LAPC4, 22Rv1 cells, and MR49F cells were maintained in RPMI-1640 media (Invitrogen Life Technologies) plus 10% heat inactivated fetal bovine serum (Invitrogen Life Technologies). VCaP cells were maintained in DMEM media with 10% FBS. MR49F cells were maintained in the presence of 10 μM ENZA. V16D and C4-2 were maintained in RPMI plus 10% charcoal stripped serum media (Thermo scientific). RWPE cells were maintained in keratinocyte-SFM (1X) media with 2.5 μg EGF human recombinant and 25 mg bovine pituitary extract supplements. All the cell lines were tested negative from mycoplasma contamination and authentication was confirmed by genomic sequencing.

## Transfection and treatments

For silencing of *SDHA* and *SDHB* genes, total $1 \times 10^6$ cells were plated in either 10-cm dish or 6-well plate as per requirement a day before transfection. Cells were transfected with 20 nM siRNAs using oligofectamine (Life technologies) reagent for next two consecutive days as per the manufacturer's instructions. RPMI plus 10% FBS (1 volume of the transfection mix) was added after 4 h of transfection each day. Cells were harvested within 24-36 h after the second transfection. Cells were treated with ENZA (10 μM), SB (10 μM), Genistein (20 μM) or STO-601 (5 μM) after second transfection before harvesting. For double-silencing experiments, $1 \times 10^6$ cells were first transfected with Hsp27, AMPK, CaMKK2, or HIF1α siRNAs using Lipofectamine RNAimax (Life technologies) as per the manufacturer's protocol at the time of seeding followed by oligofectamine transfection for SDHA or SDHB siRNAs in the similar manner as described above for next two days. For overexpression experiment, 10 μg of SDHA (Origene, RC200349) or SDHB plasmid (Addgene, 50055) was mixed with 30 μl of Xtremegene9 reagent (Sigma-Aldrich, 6365779001) or TransIT20/20 transfection reagent (Mirus, Madison, WI, USA) in 1.5 ml of optimem media and incubated at RT for 30 min. This transfection mix was added to the $1 \times 10^6$ cells plated 24 h before in RPMI plus FBS media, and cells were harvested after 48 h of incubation at 37°C. LNCaP cells were grown in CSS plus 0.1 nM R1881 for 3 days before treatment with ENZA (10 μM), ODM-201 (10 μM), and VPC-14449 (10 μM) for 6, 12, and 24 h for detecting SDHA and SDHB levels. For R1881 treatment, cells were androgen deprived by growing them for two days in RPMI plus 10% CSS media followed by stimulation with 5 nM R1881. Cells were treated with 2-5 mM DMM in RPMI plus 10% FBS media for 24 h to detect its effect on AR levels and activity. Cells were treated with ENZA (10 μM) and DMS (5 mM) or STO-609 (5 μM) for detecting changes in p-CaMKK2/ p-AMPK/p-p38/p-Hsp27 axis and AR. Different reagents and siRNAs are listed in Appendix Tables S1 and S2, respectively.

## Western blot analysis

Cells were lysed in RIPA buffer supplemented with protease inhibitors followed by protein assay performed using BCA protein assay kit (Thermo Fisher), and 40-60 μg of total protein was loaded in the SDS–PAGE followed by transfer to nitrocellulose membrane (Bio-rad Laboratories, Mississauga, ON). Membranes were blocked with 10% milk and probed for antibodies listed in Appendix Table S3.

## IHC and TMAs

We analyzed immunohistochemical expression in tissue microarrays constructed from 307 PCa cancer specimens obtained from the Vancouver Prostate Tissue Bank at the University of British Columbia (Clinical Research Ethics Board number: H09-01628). TMAs contain 70 tumors with no prior treatment, 198 patients subjected to neoadjuvant hormone therapy prior to surgery and 39 castration-resistant tumor, 15 of them (NEPC) immunoexpressed at least one NE biomarker (total $n = 307$). Only NHT samples treated for less than 6 months ($n = 130$) were used for analysis. PDX TMA was constructed from 31 PDX tumors including 14 castration-sensitive, 8 CRPCs, 4 NEPCs, and 5 dormant tumors from the *living tumor laboratory* (http://www.livingtumorlab.com/) *at Vancouver Prostate Centre*. These tumors started as adenocarcinoma morphologically and progressed to CRPC status. Four of those CRPC cases were recognized as NEPC like tumors and showed strong immunoreactivity to 2 out of 3 NE biomarkers. IHC staining was conducted using Ultra Map and Chromo Map DAB kit from Ventana Autostainer, Discovery ultrasystem. Digital images of the stained slides were acquired with SCN400 Slide Scanner (Leica Microsystems). The images were stored in the digital imaging hub (DIH; Leica Microsystems). The area of interest in the tumor images was analyzed by a pathologist, and automated digital image analysis was run for each biomarker using *Aperio Positive Pixel Count Algorithm*. Antibodies used for TMA analysis have been listed in Appendix Table S3.

## Quantitative reverse transcription PCR (qRT–PCR)

Total RNA was extracted from cultured cells using RNeasy mini kit (Qiagen, 74104) followed by cDNA synthesis using high-capacity cDNA reverse transcription kit (Thermo fisher scientific, 4368814). RT–PCR was performed on Applied Biosystems ViiA 7 from Life technologies using Taqman assay (Applied Biosystems, LS4364340) using primers listed in Appendix Table S4.

## Chromatin immunoprecipitation (ChIP) assay

LNCaP cells were grown in RPMI plus 10% FBS media for two days followed by androgen starvation for three days by replacing media with RPMI plus 10% CSS and then stimulated with 5 nM R1881 for 24 h. ChIP assay was performed using Magna ChIP A kit (Millipore, 17-610) following the manufacturer's guidelines. Briefly, cells were fixed with 1% formaldehyde followed by nuclear extraction and sonication using Covaris Sonalab 7 M220. Sheared chromatin from each sample was incubated with AR (Santacruz biotechnology, sc-816) and rabbit IgG antibodies separately for overnight followed by washing, elution and reverse cross-linking of DNA–protein complex. 5% of sheared chromatin was saved as input for the assay. DNA was further purified using spin columns provided in the kit following its instructions. Quantitative PCR was performed on the DNA purified after immunoprecipitation as well as input chromatin for *SDHA, SDHB, PSA,* and *GAPDH* probes (see probe sequences in

Appendix Table S5) using SYBR green master mix (Qiagen). PCR data were normalized for housekeeping gene *GAPDH* and compared with input.

## Electrophoretic mobility shift assay

EMSA was performed between purified recombinant AR-DNA binding domain and IRdye700–labeled oligonucleotides using Odyssey infrared EMSA kit (LI-COR, part no: 829-07910). AR-DBD was purified as reported earlier (Dalal *et al*, 2018) and was a kind gift by Dr. Kush Dalal. Oligonucleotides (listed in Appendix Table S6) were incubated with increasing amount of protein at room temperature for 30 min in the presence of 2.5 mM DTT/0.25% Tween 20, ploy (dI.dC) and then run on 6% TBE native gel and visualized on the LI-COR odyssey infrared imager.

## Extracellular flux analysis

Glycolysis stress test kit (Agilent technologies, 103020-100) was used in order to measure extracellular acidification rate (ECAR) and oxygen consumption rate (OCR). Briefly, cells were plated at 4,000 cells per well cell density in XF96-well plate pretreated with poly lysine. For silencing, cells were transfected using lipofectamine RNAimax as per manufacturer's guidelines before plating. 48 h after transfection, cells were treated with 10 μM ENZA for 12 h. On the day of experiment, media were replaced with phenol-red free Seahorse XF base medium (Agilent, 103335-100) containing 2 mM L-glutamine. 10 mM glucose, 2 μM oligomycin, and 50 mM 2-DG were added in ports A, B, and C on indicated times. Data were normalized to cell number measured by crystal violet assay. For crystal violet assay, cells were sequentially incubated with 0.1% glutaraldehyde and crystal violet for 5-15 min each. After washing excess stain, plate was dried. Finally, 100 μl of Sorenson's solution was added to each well and absorbance taken at 570 nm. Data analysis was performed using Wave Desktop 2.6 software from Agilent technologies.

## Real-time cell proliferation and caspase assay

LNCaP cells were transfected with different siRNAs using Lipofectamine RNAimax and simultaneously plated in 96 well plate at the cell density of 4000 cells in 100 μl per well. Cells were treated with 10 μM ENZA after 24 h of transfection and incubated in the IncuCyte® S3 Live-cell analysis system for real time image acquisition. Cell proliferation data (phase) was quantified using the IncuCyte® basic analyzer.

## Calcium flux analysis

10,000 cells were plated in Lab Tek II chamber slide system (Thermo Fisher, 154534) in 500 μl RPMI plus FBS media. Next day, media were replaced with serum-free RPMI media and cells were treated with 2 μM thapsigargin, 5 mM DMS or 2 mM DMM for 1 h and 10 μM ENZA for 1 and 6 h, respectively. It was followed by 30 min incubation with 5 μM Fluo4-AM probe in HBBS media containing 1mM EDTA at 37°C along with different treatments (DMS, DMM, ENZA). Imaging was performed using Leica 780 confocal microscope.

## Cell viability assay

Cell viability was measured using cell counting kit 8 from Dojindo Molecular Technologies Inc. (CK04-05) as per its guidelines. Briefly, 3,000-5,000 cells per well were plated in 96 well plate in RPMI plus 10% FBS media one day before treatment. Next day, cells were treated with ENZA and then incubated at 37°C for 72 h. 10 μl of dye provided by the kit was added to each well and incubated for 2 h at 37°C before recording absorbance on 450 nm. For silencing and overexpression experiments, cells were directly plated in 96-well plate with lipofectamine RNAimax or xtremegene9 reagents, respectively.

## *In vivo* study

In vivo study with LNCaP xenograft was performed as described before (Nappi *et al*, 2020). Animal study was approved by the Animal Care Committee of the University of British Columbia (protocol number A18-0118) and performed as per guidelines of Canadian Council on Animal Care and appropriate institutional certification. Briefly, $2 \times 10^6$ LNCaP cells were inoculated s.c. into the left and right flanks of 6- to 8-week-old male athymic nude mice ($n = 10$). Once the PSA reached 50 ng/ml or the tumor volume 300 mm$^3$, the animals were surgically castrated and then subsequently randomized to receive vehicle or 10 mg/kg IVM orally three times a week. Mice were sacrificed after 12 weeks of treatment and tumors extracted. For acute castration, mice were sacrificed 3 days after castration ($n = 5$). Pre-castration tumors were extracted from mice which did not undergo castration ($n = 5$). CRPC tumors were harvested 3 months after castration ($n = 5$). AR and p-Hsp27 levels were determined on these tissues using Western blotting using antibodies as mentioned in Appendix Table S3.

## SDH activity assay

Activity assay was performed as per manufacturer's guidelines (Sigma MAK197-1KT). Briefly, cells treated under different conditions were harvested and washed once with PBC. Pellets were resuspended in 3-4 volumes of PBS and homogenized by pipetting up and down after adding 1/10th volume of 10X detergent provided in the kit. After 30 min incubation, cells were centrifuged, and BCA assay was performed on the supernatant. 50-100 μg of total protein was used for performing activity assay.

## Succinate metabolite analysis

Cell pellets (10-20 mg) were extracted with 40 μl 0.1 M triethylamine (TEA) and vortexing, followed by addition of 2 μl of deuterated succinic acid (2,2,3,3-d$_4$, IS) and 120 μl acetonitrile and further vortexing and subsequent sonication for 5 min. Remaining solids were sedimented (20,000 *g*, 5 min), and the supernatant was vacuum dried with a centrifugal evaporator (Centrivap). The residue was taken up in 50 μl water, sonicated for 5 min, centrifuged (20,000 *g*, 5 min), and supernatants were transferred to LC vials. Analysis was carried out with a Waters Acquity UPLC Separations Module coupled to a Waters Quattro Premier XE Mass Spectrometer. Separations were with a 2.1x100 mm BEH 1.7 μM C18 column, mobile phase 2 mM ammonium formate/0.1% formic acid in water (A) and similarly in methanol (B) (gradient: 0-0.2 min, 3% B; 0.2-

1 min, 3-30% B; 1-2 min, 30-97% B; 2-2.5 min 97% B; 2.5-2.6 min, 97-3% B; 5 min run length). All data were collected in ES (-) by multireaction monitoring with the following instrument parameters: capillary 3 kV; extractor/RF 3/0.1 V; source/desolvation temperatures 120/300°C; cone/desolvation $N_2$ flow 50/1000 l/h; collision Ar gas $8.2e^{-3}$ mbar. Masses used and cone/collision voltages were as follows: succinic acid m/z117 > 73, 17V/12V; lactic acid m/z89 > 43, 28V/9V; d4-succinic acid m/z121 > 77, 17V/14V. Five calibration samples (matrix free with equivalent IS) ranging to 50 μM were prepared for reference. Post spiked samples indicated matrix effects were minimal, allowing use of d4 succinic IS for all analytes. Raw data processing was carried out using Quanlynx (Waters) and normalized against pellet weights.

## Tandem mass tagging assay

Castrate-sensitive LNCaP cells were grown in CSS + 0.1 nM DHT for three days followed by treatment with 10 μM ENZA for AR blockade for 6, 12 and 24 h. Protein lysates were labeled using a 10-plex Tandem Mass Tag (TMT) assay and subjected to mass spectrometry (MS). Protein abundance was measured, and significant changes were analyzed using ANOVA with a permutation based false detection rate (FDR) of 0.05 and 250 randomizations. Functional pathways were analyzed.

## AR transactivation assay

For luciferase assays, LNCaP cells were seeded in 24 well plates in RPMI 1640 medium with 5% FBS for 24 h, followed by transfection with 200 ng of reporter plasmid ARR3tk-luciferase or a construct driven by three repeats of an AR-V7–specific promoter element of the ubiquitin-conjugating enzyme E2C (*UBE2C*) gene along with 0.6 μl/well TransIT2020 transfection reagent (Mirus, Madison, WI, USA) and 50 μl/well Optimem complete medium (Gibco, Langley, OK, USA) for 24 h at 37°C. Cells were then treated with ENZA (10 μM), DMM (2 mM) or IVM (2.5 and 5 μM) for 12 h. Luciferase activity was measured as previously reported (Xu *et al*, 2006). For AR transactivation assay with SDH silenced cells, cells were transfected with siRNAs at the time of seeding using Lipofectamine RNAimax as per manufacturer's guidelines followed by transfection with ARR3tk-luciferase plasmid next day. For overexpression experiments, cells were transfected with 10 μg of SDHA/SDHB plasmid along with ARR3tk-luciferase plasmid. 48 h after transfection, cells were treated with 10 μM ENZA for 12 h before measuring luciferase activity. Data were normalized with Renilla activity or total protein.

## ATP measurement

4,000 cells per well were plated in 96-well plate with siScr (scrambled), SDHA and SDHB siRNAs as a complex with Lipofectamine RNAimax. After 48 h of transfection, cells were treated with 10 μM ENZA for 12 h. ATP assay was performed as per manufacturer's guidelines (Abcam, ab113849).

## Statistical analysis

Statistical significance was assessed using a two-tailed unpaired Student's *t*-test when two groups were compared whereas one-way

### The paper explained

#### Problem

Development of treatment resistance is a common feature of solid tumor malignancies and the cause of most cancer deaths. As the main driver of prostate cancer energetics and growth, androgen receptor (AR) pathway inhibition (ARPI) is standard first-line therapy for advanced prostate cancer. Despite frequent and durable responses, treatment resistance emerges under the selective pressures of ARPI via adaptive survival pathways activation, mutagenesis, and clonal evolution, a complexity amplified by multi-clonal and inter-patient genomic heterogeneity. Functional redundancy, which includes metabolic remodeling, fuels the plasticity that cancers display, enabling rapid response and adaption to external stressors and targeted therapies.

#### Results

We explored early stress responses that orchestrate shifts in energetics for metabolic homeostasis after acute ARPI that support survival under treatment stress, key for subsequent emergence of resistance. We identified that the AR, via AREs, directly regulated expression of SDH catalytic subunits, an integral enzyme of energy metabolism. Acute ARPI inhibited SDH activity, leading to intracellular succinate accumulation that triggered $Ca^{2+}$ release from internal stores and subsequent activation of p-CAMKK2/p-AMPK/p-p38 axis, culminating in increased levels of the AR co-chaperone, p-Hsp27, to enhance AR protein stability and signaling. Co-targeting p-Hsp27 with ARPI abrogated this succinate-mediated adaptive loop and enhanced prostate cancer sensitivity to ARPI.

#### Impact

This study defines how AR directly regulates energy metabolism in prostate cancer and how cells acutely reprogram energetics to support survival and adaption in response to acute treatment stress. It further defines succinate as an oncometabolite that triggers stress signaling for AR reactivation via p-Hsp27, uncovering a potential therapeutic strategy involving co-targeted inhibition of Hsp27 with ARPI to abrogate this metabolic adaptive response.

analysis of variance (ANOVA) followed by Tukey's correction for three or more groups. GraphPad Prism 6 software was used to calculate the statistical significance. The threshold of statistical significance was set at $*P < 0.05$, $**P < 0.01$, $***P < 0.001$, $****P < 0.0001$.

## GSEA analysis

GSEA v4.0.3 Software from the Broad Institute (Cambridge, MA, USA) (Subramanian *et al*, 2005) was used for pathway analysis of patient gene expression data (Labrecque *et al*, 2019) classified into "ARhighNEnegative" and "ARnegativeNEpositive" to identify differentially expressed genes in the Molecular Signature database, v7.2 which was run in classic mode. Pathways enriched with nominal $P < 0.05$ and false discovery rate < 0.25 were considered to be significant.

# Data availability

Additional data are available online in the Expanded View and Appendix. This study includes no data deposited in external repositories.

**Expanded View** for this article is available online.

## Acknowledgements

This study was supported by Terry Fox New Frontiers Program Project Grant # 1062. Authors would like to acknowledge Dr. Dwaipayan Ganguli, Dr. Kush Dalal, Dr. Noushin Nabavi, Jeffery Leong, Ivan Asmaro, Pan Hsu Lin, Mary Bowden, and Virginia Yago for providing reagents, technical and intellectual help.

## Author contributions

Supervision: MG; Experiment design: NS, EB, FZ, SPS, and MG; Experiments: NS, EB, LF, KN, PB, SPS, AG, CM, HA, and NN; Data analysis and scientific discussion: NS, MG, EB, LF, FZ, SPS, LN, YW, CC, and PS; Writing manuscript: NS; Proofreading manuscript: EB, FZ, SPS, PH.B., and M.G.

## Conflict of interest

The University of British Columbia has granted patents on the Hsp27 antisense OGX-427. A patent application (62/756,707) has been submitted applications on ivermectin analogues for the treatment of cancer, listing Dr. Martin E. Gleave as inventor.

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
