## [Review Process File · EMBO Molecular Medicine]

Androgen Receptor (AR) antagonism triggers acute succinate-mediated adaptive responses to reactivate AR signaling

Neetu Saxena, Eliana Beraldi, Ladan Fazli, Syam Prakash Somasekharan, Hans Adomat, Fan Zhang, Chidi Molokwu, Anna Gleave, Lucia Nappi, Kimberly Nguyen, Pavn Brar, Nicholas Nikesitch, Yuzhuo Wang, Colin Collins, Poul Sorensen, and Martin Gleave

DOI: [10.15252/emmm.202013427](https://doi.org/10.15252/emmm.202013427)

Corresponding authors: [Martin Gleave \(m.gleave@ubc.ca\)](mailto:m.gleave@ubc.ca)

Review Timeline:

Submission Date:	9th Sep 20
Editorial Decision:	29th Sep 20
Revision Received:	22nd Dec 20
Editorial Decision:	25th Jan 21
Revision Received:	5th Feb 21
Accepted:	9th Feb 21

Editor: *Lise Roth*

Transaction Report:

29th Sep 2020

Dear Dr. Gleave,

Thank you for the submission of your manuscript to EMBO Molecular Medicine. We have now received feedback from the three reviewers who agreed to evaluate your manuscript. As you will see from the reports below, the referees acknowledge the interest of the study and are overall supporting publication of your work pending appropriate revisions.

Addressing the reviewers' concerns in full will be necessary for further considering the manuscript in our journal, with the exception of the *in vivo* experiments mentioned by referee #3. Indeed, upon consultation, the three referees agreed that additional xenograft experiments with cells overexpressing SDHA/B and treated with Enzalutamide would strongly benefit the manuscript, but due to the amount of time and work required, would not be necessary for acceptance of a revised manuscript.

Acceptance of the manuscript will entail a second round of review. EMBO Molecular Medicine encourages a single round of revision only and therefore, acceptance or rejection of the manuscript will depend on the completeness of your responses included in the next, final version of the manuscript. For this reason, and to save you from any frustrations in the end, I would strongly advise against returning an incomplete revision.

When submitting your revised manuscript, please carefully review the instructions that follow below. Failure to include requested items will delay the evaluation of your revision:

- 1) A .docx formatted version of the manuscript text (including legends for main figures, EV figures and tables). Please make sure that the changes are highlighted to be clearly visible.
- 2) Individual production quality figure files as .eps, .tif, .jpg (one file per figure).
- 3) A .docx formatted letter INCLUDING the reviewers' reports and your detailed point-by-point responses to their comments. As part of the EMBO Press transparent editorial process, the point-by-point response is part of the Review Process File (RPF), which will be published alongside your paper.
- 4) A complete author checklist, which you can download from our author guidelines (<https://www.embopress.org/page/journal/17574684/authorguide#submissionofrevisions>). Please insert information in the checklist that is also reflected in the manuscript. The completed author checklist will also be part of the RPF.
- 5) Before submitting your revision, primary datasets produced in this study need to be deposited in an appropriate public database (see <https://www.embopress.org/page/journal/17574684/authorguide#dataavailability>). Please remember to provide a reviewer password if the datasets are not yet public. The accession numbers and database should be listed in a formal "Data Availability" section (placed after Materials & Method). Please note that the Data Availability Section is restricted to

new primary data that are part of this study.

6) We would also encourage you to include the source data for figure panels that show essential data. Numerical data should be provided as individual .xls or .csv files (including a tab describing the data). For blots or microscopy, uncropped images should be submitted (using a zip archive if multiple images need to be supplied for one panel). Additional information on source data and instruction on how to label the files are available at .

7) Our journal encourages inclusion of *data citations in the reference list* to directly cite datasets that were re-used and obtained from public databases. Data citations in the article text are distinct from normal bibliographical citations and should directly link to the database records from which the data can be accessed. In the main text, data citations are formatted as follows: "Data ref: Smith et al, 2001" or "Data ref: NCBI Sequence Read Archive PRJNA342805, 2017". In the Reference list, data citations must be labeled with "[DATASET]". A data reference must provide the database name, accession number/identifiers and a resolvable link to the landing page from which the data can be accessed at the end of the reference. Further instructions are available at .

8) We replaced Supplementary Information with Expanded View (EV) Figures and Tables that are collapsible/expandable online. A maximum of 5 EV Figures can be typeset. EV Figures should be cited as 'Figure EV1, Figure EV2" etc... in the text and their respective legends should be included in the main text after the legends of regular figures.

- Additional Tables/Datasets should be labeled and referred to as Table EV1, Dataset EV1, etc. Legends have to be provided in a separate tab in case of .xls files. Alternatively, the legend can be supplied as a separate text file (README) and zipped together with the Table/Dataset file. See detailed instructions here:

9) The paper explained: EMBO Molecular Medicine articles are accompanied by a summary of the articles to emphasize the major findings in the paper and their medical implications for the non-specialist reader. Please provide a draft summary of your article highlighting

10) For more information: There is space at the end of each article to list relevant web links for further consultation by our readers. Could you identify some relevant ones and provide such information as well? Some examples are patient associations, relevant databases, OMIM/proteins/genes links, author's websites, etc...

11) Every published paper now includes a 'Synopsis' to further enhance discoverability. Synopses are displayed on the journal webpage and are freely accessible to all readers. They include a short stand first (maximum of 300 characters, including space) as well as 2-5 one-sentences bullet points that summarizes the paper. Please write the bullet points to summarize the key NEW findings. They should be designed to be complementary to the abstract - i.e. not repeat the same text. We encourage inclusion of key acronyms and quantitative information (maximum of 30 words / bullet point). Please use the passive voice. Please attach these in a separate file or send them by email, we will incorporate them accordingly.

Please also suggest a striking image or visual abstract to illustrate your article. If you do please provide a png file 550 px-wide x 400-px high.

12) As part of the EMBO Publications transparent editorial process initiative (see our Editorial at <http://embomolmed.embopress.org/content/2/9/329>), EMBO Molecular Medicine will publish online a Review Process File (RPF) to accompany accepted manuscripts.

In the event of acceptance, this file will be published in conjunction with your paper and will include the anonymous referee reports, your point-by-point response and all pertinent correspondence relating to the manuscript. Let us know whether you agree with the publication of the RPF and as here, if you want to remove or not any figures from it prior to publication.

I look forward to receiving your revised manuscript.

Yours sincerely,

Lise Roth

Lise Roth, PhD
Editor
EMBO Molecular Medicine

***** Reviewer's comments *****

Referee #1 (Comments on Novelty/Model System for Author):

They apply state of the art methodologies to investigate the mechanisms underlying the problem of castration-resistant prostate cancer and they find a potential therapeutic solution.

Referee #1 (Remarks for Author):

The manuscript by Sasexa et al., is an interesting piece of work that highlights the role of metabolism in the activation of the gene expression programs and signaling pathways that reactivate the androgen receptor (AR) after treatments aimed at its inhibition in prostate cancer; ultimately resulting in cancer resistant to treatment. They provide evidence to support that the inhibition of the androgen receptor pathway (ARPI) triggers transcriptional down-regulation of SDH genes, the accumulation of cellular succinate and the long-term resurgence of the AR by stabilization of the protein. With loss and gain of function experiments, they correlate low SDH levels with cellular reprogramming to an enhanced glycolysis, the activation by phosphorylation of p38, AMPK, CaMKK2 and of Hsp27, which is the co-chaperone of the AR, responsible for the resurgence of the AR in response to ARPI. Moreover, they present data to suggest that the cellular accumulation of succinate in response to ARPI could act as the signaling molecule that exerts the increase in cytoplasmic Ca²⁺ that activates the CaMKK2/AMPK/p38/Hsp27 signaling pathway that restores the AR in prostate cancer cells. Importantly, these results are supported by clinical data of protein expression in prostate tissue microarrays that show the down-regulation of SDHA and the concurrent upregulation of phospho-CaMKK2, phospho-AMPK and phospho-Hsp27 in response to androgen deprivation therapies in cancer patients. Remarkably, they also document in prostate cancer cells and xenografts that the reactivation of the AR in response to androgen deprivation therapies could be prevented by co-treatment with ivermectine, an inhibitor that prevents the phosphorylation of Hsp27, and hence the stabilization of the AR, hampering the proliferation of cancer cells. Overall, the work is very solid, with the appropriate controls and very well presented. I have minor comments that might improve the presentation of the final version of the manuscript.

Minor points

1. Fig. 3e; The bar graph of the quantification needs the incorporation of additional data and

statistical evaluation of the results obtained.

2. Can the participation of G-protein coupled receptors be excluded in the Ca²⁺ signal mediated by succinate? Is the surge of succinate inhibiting the flux of autophagy? What is the intracellular concentration of succinate achieved by incubation of the cells with ENZA or CSS (Fig. 1d)? Is the succinate-Ca²⁺ response specific of prostate cancer cells?

3. In Figure 6, a new panel incorporating the effect of the treatments in the tumor size of the xenografts is required.

4. On page 17, the reference by Zoubeidi, Zardan et al Cancer Res (2007) 67:10455-10465 is listed twice.

Referee #2 (Remarks for Author):

In this manuscript Saxena et al. demonstrate direct regulation of SDHA/B by the AR signalling axis in prostate cancer cells. They further uncovered a feedback regulation pathway whereby SDHA/B appear to down-regulate AR protein expression itself by working as a tumor suppressor. They next showed ARPI such as ENZA induced SDH inhibition causes succinate build up leading to activation of AR cochaperone Hsp27 via AMPK/CAMKK2/p38 pathway. Overall their research describes a early mechanisms by which resistance to AR inhibitors can potentially emerge in CRPC. Several conclusions are supported by a robust set of experiments thus underscoring the importance of their findings. The manuscript in general is well written but will benefit from more information provided on experiments in figure legends such as time points, cell lines etc.

Comments

1. Fig 1d: showing ARPI induced succinate upregulation contrasts from Massie et al. (see PMC315295), that reported levels of succinate remain unchanged by androgen manipulation. Authors should test how the androgens affect the levels of succinate in their cell lines and provide a comparison/discussion.
2. Fig 2a: loading control of LAPC-4 makes it difficult to appreciate the results in this cell line and should be repeated. This appears to be a general problem with almost all Western blots where Vinculin was used as internal control.
3. Fig 2b: only shows effect of SDHB knockdown on AR levels, how does SDHA knockdown (kd) affect AR expression/activity? Does SDHA silencing also restore AR expression? It should also be tested whether SDHA/B kd in ENZA resistance cells produces a distinct AR protein expression kinetics compared to ARPI sensitive cells.
4. Fig 2c/d: authors should also test what is the combined effect of SDHA and B kd on AR levels/activity.
5. Fig. 3e: authors should use additional control (vehicle) for Enzalutamide in this experiment. This is an important aspect of the paper to demonstrate effect of ENZA or p38 and the phosphorylation of its downstream target Hsp27.
6. P38 is a pro-survival kinase and its levels can increase upon general cellular stresses and not specifically in response to ARPI. To test this, authors should test what happens to ca²⁺/CAMKK2/AMPK/p38/p-Hsp27 axis in cell lines that ectopically express SDHA and B?
7. It is unclear why SDHA kd affects SDHB levels (fig 3e). Authors may consider using 2-3 different siRNAs.
8. It is shown that ARPI-induced repression of SDH activity leads to adaptive increases in AR protein levels, it is unclear as to why this increase is only observed at earlier time points following ENZA (fig 2B) and not at later time points.
9. Authors suggest that "castration-resistant V16D and ENZA-resistant MR49F cell lines which maintain AR activity independent of androgen availability demonstrate more SDH activity compared

to castration sensitive LNCaP cells". It is unclear what happens to the SDH-AR axis in these resistant cell lines? Would increased SDH activity not be sufficient to clear succinate and eventually block AR activation- is this not counterintuitive?

10. Data show "SDH-repressed cells demonstrated higher bioenergetic profiles by primarily increasing glycolysis (up to 100% increase)" how would they explain increased SDH activity observed in CRPC cell lines (Supp fig 3e). Does that mean CRPC is less reliant on glycolysis?

11. Inconsistent: fig 3e shows p-Hsp7 levels are enhanced following SDHA or B kd; however, in figure 4c SDHA/SDHB kd doesn't seem to decrease p-Hsp27 levels, please clarify.

13. Authors show (supp fig 5B) lower AR levels/pHsp27 levels in NEPC, it would be interesting to test whether this is due to upregulation of SDH activation independent of AR axis in such tumors. Authors should also explain increased SDHA activity observed in CRPC patients (fig 5c)

Referee #3 (Comments on Novelty/Model System for Author):

The review of the article shows that the manuscript meets all required qualities for Embo Mol Med publication level. Technical, novelty and medical aspect are strong enough to go ahead with this study.

Referee #3 (Remarks for Author):

Here, Saxena et al. report the crucial role of succinate dehydrogenase (SDH) activity in reactivating AR signaling following an Androgen Receptor (AR) pathway inhibition (ARPI) strategy. Authors show that TCA cycle, and especially SDH A and B subunits, are controlled by AR through a "classical" transcriptional regulation. Reciprocally, authors show that inhibition of SDH results in AR protein accumulation and activity underlying a potential role in ARPI adaptation. Using mechanistic approaches, the authors demonstrated that succinate accumulation, resulting from SDH inhibition induced by ARPI, activated a calcium-dependent axis sustained by CAMKK2, AMPK, p38-MAPK and HSP27 to promote AR stabilization and to bypass AR inhibition. These findings have been correlated with human TMA investigations. Finally, the downstream target HSP27 inhibition, using the small inhibitor Ivermectin, has been challenged using a preclinical xenograft model. The experiments are well designed and performed; the statistical analysis meets sufficiency requirements. The manuscript is well written, and the results are extremely relevant as they proposed an early metabolic adaptation of PCa tumor cell to Androgen Deprivation Therapy. Biological questions are rigorously addressed using a wide range of experimental approaches that strengthened research rational and experimental workflow.

Although the manuscript is consistent and well conducted, some issues should be addressed to make the study more robust and more attractive for non-specialist readers.

Major issues:

- 1) Paper rational is based on the central role of SDHA/B in acute ARPI resistance mechanism. Although the authors provide really clear-cut in vivo experiments using Ivermectin to target HSP27, pre-clinical investigation missed in vivo targeting of SDHA/B to validate succinate level as a critical crossroad to manage ARPI resistance. Thus, authors should test xenograft models using LNCaP cells or another cell line that overexpress SDHA/B and challenge Enzalutamide effect.
- 2) Extrapolation of data regarding SDHA/B status and Androgen Regulated Genes (ARG) signature should be investigated using public dataset in order to correlate SDHA/B expression with ARG.

GSEA analysis should be an option.

3) Cell culture model limitations come from population clonality and cell specific response. Thus, authors should determine SDH activity and succinate accumulation (as depicted in Figure 1c/d) in others AR-positive prostate cancer cell lines as these experiments have been performed only in LNCaP cells. Authors should include at least similar experiments using LAPC4 cell line.

Minor issues:

1) Figure 1F : Authors should provide statistical analysis for the ChIP assay. Y-axis legend should be corrected and mentioned "relative enrichment analysis/quantification" instead of "Relative mRNA levels"

2) Figure S1a : SDHA results obtained with Mass Spectrometry approach exhibit an increased accumulation after 24 hours Enzalutamide exposition. Do the authors have an explanation for this result?

3) ARE binding site is characterized by an IR3 binding-site; hexamers half-site should be indicated by uppercase letters in Fig S1F.

4) Figure 5a,b,c: The significance NHT, abbreviation corresponding to "neoadjuvant-treated patients", needs to be indicated in the text legend.

5) Figure 3c, Figure 6c,d : Some of real time cell proliferation plots miss the control curve. Homogenization of the curve color charts between each plot should facilitate analysis. Statistical analysis should be included at least for the final time point.

6) A recent paper pointed out the key role of succinate accumulation in high-grade prostate cancer samples. This study provides important complementary results that strengthen the present article message. This article should be added in the manuscript and discussion. Nat Commun. 2020 Mar 20;11(1):1487.doi: 10.1038/s41467-020-15237-5. PMID: 32198407

REFREE 1:

They apply state of the art methodologies to investigate the mechanisms underlying the problem of castration-resistant prostate cancer and they find a potential therapeutic solution.

Comments for the Authors

The manuscript by Sasexa et al., is an interesting piece of work that highlights the role of metabolism in the activation of the gene expression programs and signaling pathways that reactivate the androgen receptor (AR) after treatments aimed at its inhibition in prostate cancer; ultimately resulting in cancer resistant to treatment. They provide evidence to support that the inhibition of the androgen receptor pathway (ARPI) triggers transcriptional down-regulation of SDH genes, the accumulation of cellular succinate and the long-term resurgence of the AR by stabilization of the protein. With loss and gain of function experiments, they correlate low SDH levels with cellular reprogramming to an enhanced glycolysis, the activation by phosphorylation of p38, AMPK, CaMKK2 and of Hsp27, which is the co-chaperone of the AR, responsible for the resurgence of the AR in response to ARPI. Moreover, they present data to suggest that the cellular accumulation of succinate in response to ARPI could act as the signaling molecule that exerts the increase in cytoplasmic Ca²⁺ that activates the CaMKK2/AMPK/p38/Hsp27 signaling pathway that restores the AR in prostate cancer cells. Importantly, these results are supported by clinical data of protein expression in prostate tissue microarrays that show the down-regulation of SDHA and the concurrent upregulation of phospho-CaMKK2, phospho-AMPK and phospho-Hsp27 in response to androgen deprivation therapies in cancer patients. Remarkably, they also document in prostate cancer cells and xenografts that the reactivation of the AR in response to androgen deprivation therapies could be prevented by co-treatment with ivermectin, an inhibitor that prevents the phosphorylation of Hsp27, and hence the stabilization of the AR, hampering the proliferation of cancer cells. Overall, the work is very solid, with the appropriate controls and very well presented. I have minor comments that might improve the presentation of the final version of the manuscript.

Response: We thank the reviewer for positive feedback and valuable suggestions.

Minor points:

1. Fig. 3e; The bar graph of the quantification needs the incorporation of additional data and statistical evaluation of the results obtained.

Response: As per the reviewer's suggestion, we collected additional data and performed statistical analysis of the results obtained that are included in the Fig. 3E (right panel).

2. Can the participation of G-protein coupled receptors be excluded in the Ca²⁺ signal mediated by succinate? Is the surge of succinate inhibiting the flux of autophagy? What is the intracellular

concentration of succinate achieved by incubation of the cells with ENZA or CSS (Fig. 1d)? Is the succinate-Ca²⁺ response specific of prostate cancer cells?

Response:

- While benign prostate tissue has not been shown to express SUCNR1 receptor as determined from the Human Protein Atlas (RNA expression data from Consensus dataset in panel “A”), prostate cancer cells can express this receptor (RNA expression from TCGA dataset in panel “B”), therefore we do not rule out a possibility of its involvement in succinate mediated Ca²⁺ signaling.

A. SUCNR1 distribution in benign tissues

B. SUCNR1 distribution in different cancer types

Fig. 1: RNA expression of SUCNR1 in prostate benign (A) and cancer (B) cells.

- We have reported that acute ARPI stress increases autophagy (Zhang *et al*, 2014). Previously Kumar *et al* (Kumar *et al*, 2019) showed that ethanol induces mitochondrial dysfunction and oxidative stress, inhibits succinate oxidation, dysregulates protein synthesis and induces autophagy in skeletal muscles. In addition, succinate stabilizes HIF1alpha, which can upregulate autophagy (Ajdukovic, 2016; Glick *et al*, 2010; Yu *et al*, 2019). Furthermore, succinate accumulation during myocardial ischemia/reperfusion injury increases mitochondrial ROS sensitizing opening of mitochondrial permeability transition pore, which promotes autophagy (Zhou *et al*, 2019). Therefore, based on this evidence, we do not expect surge of succinate to inhibit autophagy flux.
- The intracellular concentration of succinate in LNCaP cells is 3-6 μM per million cells under untreated conditions which increased up to 10-12 μM per million cells after incubation with ENZA or CSS.
- Succinate- Ca^{2+} response is not specific to prostate cancer cells as previous studies have shown similar observations. Succinate is known to induce calcium mobilization through SUCNR1, a G-protein coupled receptor (Aguiar *et al*, 2010; Mills & O'Neill, 2014). Activation of SUCNR1 by succinate activates phospholipase C (PLC), which cleaves phosphatidylinositol 4,5-biphosphate (PIP2) to inositol triphosphate (IP3) and diacylglycerol (DAG). IP3 further leads to Ca^{2+} releases into cytoplasm.

3. In Figure 6, a new panel incorporating the effect of the treatments in the tumor size of the xenografts is required.

Response: A new panel (E) has been included in figure 6 showing LNCaP xenograft tumor volumes and serum PSA values after treatment with ivermectin vs vehicle (Nappi *et al*, JCI, 2020), and cited in the figure legends.

4. On page 17, the reference by Zoubeydi, Zardan *et al* Cancer Res (2007) 67:10455-10465 is listed twice.

The duplicate entry has been removed.

REFREE 2:

In this manuscript Saxena et al. demonstrate direct regulation of SDHA/B by the AR signalling axis in prostate cancer cells. They further uncovered a feedback regulation pathway whereby SDHA/B appear to down-regulate AR protein expression itself by working as a tumor suppressor. They next showed ARPI such as ENZA induced SDH inhibition causes succinate build up leading to activation of AR cochaperone Hsp27 via AMPK/CAMKK2/p38 pathway. Overall their research describes a early mechanisms by which resistance to AR inhibitors can potentially emerge in CRPC. Several conclusions are supported by a robust set of experiments thus underscoring the importance of their findings. The manuscript in general is well written but will benefit from more information provided on experiments in figure legends such as time points, cell lines etc.

Response: We thank the reviewer for this insightful feedback, the responses to which have improved our manuscript. We believe we have now reasonably addressed the concerns of the reviewer, as provided below.

Comments for the Authors:

1. Fig 1d: showing ARPI induced succinate upregulation contrasts from Massie et al. (see PMC3155295), that reported levels of succinate remain unchanged by androgen manipulation. Authors should test how the androgens affect the levels of succinate in their cell lines and provide a comparison/discussion.

Response: While Massie et al tested succinate levels under agonistic condition (androgen treatment), our study focuses on antagonistic responses (Enzalutamide, CSS). As AR fuels different pathways of metabolism including glycolysis, mitochondrial respiration, and lipid metabolism, androgens will maintain metabolic flux, which makes it harder to detect differences in succinate levels. In contrast, AR antagonism may block the flux due to reduced SDH activity leading to accumulation of succinate, which can be quantified. Additionally, we tested succinate after acute ARPI treatment (12-24 h) whereas in Massie *et al* cells were treated for longer durations (up to 3 days) after which cells adapted with time. In fact, we observed 70 and 44% % reduction in succinate buildup once 1 nM R1881 was added for 24 h after CSS starvation in LNCaP or LAPC4 cells, respectively (please see the figure below) suggesting that it is easier to identify changes in succinate levels after acute treatment before cells get to adapt for changing conditions.

Fig. 2: Intracellular succinate measured in LNCaP and LAPC4 cells using LC-MS. *, $p < 0.05$; **, $p < 0.01$.

2. Fig 2a: loading control of LAPC-4 makes it difficult to appreciate the results in this cell line and should be repeated. This appears to be a general problem with almost all Western blots where Vinculin was used as internal control.

Response: We agree with the reviewer that the quality of the loading control in some figures was not optimal therefore we replaced vinculin with actin/GAPDH where feasible, or re-probed membranes with higher concentration of vinculin antibody to improve the quality of figure. Loading controls were changed in the following figures:

Fig. 2A: Vinculin replaced by GAPDH for LAPC4 samples.

Fig. 2E: Membrane was re-probed with GAPDH antibody for SDHB overexpression samples.

Fig3A. Vin replaced with actin for si SDHA samples.

Fig 3B: Vin replaced with GAPDH.

Fig. 3E. Membrane re-probed with vinculin antibody.

Fig. 4E. Membrane re-probed with GAPDH antibody therefore Vin replaced with GAPDH.

Fig. EV2F: Vinculin replaced with GAPDH antibody.

Fig. EV2H: Membrane re-probed with Vinculin antibody.

Fig. EV3E: Vin replaced with GAPDH.

3. Fig 2b: only shows effect of SDHB knockdown on AR levels, how does SDHA knockdown (kd) affect AR expression/activity? Does SDHA silencing also restore AR expression? It should also be tested whether SDHA/B kd in ENZA resistance cells produces a distinct AR protein expression kinetics compared to ARPI sensitive cells.

Response: As shown in figures 2A, 3A, 3E, 4C, 4F, and 4G, silencing of both SDHA and SDHB subunits increase AR protein levels and activity. Silencing of either SDHA or SDHB subunit led to stabilization of AR protein as shown in figure 3A. Similarly, figures 2C and 2D confirm that silencing of both subunits upregulate AR activity. Since SDHA and SDHB are both integral for formation of SDH complex, silencing of one subunit affects assembly and activity of the whole complex; therefore silencing of SDHB would result in SDH complex destabilization and we do not think it is necessary to repeat AR kinetics as shown in fig. 2B with SDHA silencing.

We have shown in figure 2a that ENZA-resistant cell lines MR49F and 22Rv1 show increased AR levels in a similar manner to castration-sensitive LNCaP and LAPC4 cell lines. However, as MR49F and 22Rv1 cells are already adapted and resistant to ENZA, it is not possible to appreciate the effect of ENZA on the AR protein expression kinetics. In fact, MR49F cells have F877L agonizing mutation in the ligand binding domain of AR, which makes ENZA an agonist for these cells (Coleman *et al*, 2016). On the other hand, 22Rv1 cells express AR variant 7 which lacks the ligand binding domain making them insensitive to ENZA (Wadosky & Koochekpour, 2017). Our study focuses on early adaptive response against antagonism when cells are still dependent on AR signaling, rather than on genomically altered cells that are resistant to ARPI.

4. Fig 2c/d: authors should also test what is the combined effect of SDHA and B kd on AR levels/activity.

Response: As the reviewer suggested, we tested AR protein and activity levels after co-silencing of SDHA and SDHB subunits in LNCaP cells. Indeed, co-silencing further increased AR protein levels (Fig. A) and AR activity (Fig. B) compared to SDHA/SDHB silencing alone. Co-silencing likely destabilizes SDH complex more efficiently and rapidly compared to mono-silencing, thereby further increasing AR protein and activity. **, $p < 0.01$.

5. Fig. 3e: authors should use additional control (vehicle) for Enzalutamide in this experiment. This is an important aspect of the paper to demonstrate effect of ENZA or p38 and the phosphorylation of its downstream target Hsp27.

Response: We agree with the reviewer on the importance of this aspect and therefore have included a new panel (panel C in the figure EV3) demonstrating increased phosphorylation of p38 and Hsp27 after 12 hr of enzalutamide treatment in LNCaP cells (please see the figure below). This finding was previously reported in a recent publication from our group (Nappi *et al*, 2020). In addition, a recent study by Ware *et al* also demonstrates that enzalutamide-resistant cells derived from LNCaP, CS2, and LN95 cell lines increase p38 phosphorylation as an adaptive response to treatment stress (<https://doi.org/10.1101/2020.04.22.050385>).

6. P38 is a pro-survival kinase and its levels can increase upon general cellular stresses and not specifically in response to ARPI. To test this, authors should test what happens to ca²⁺/CAMKK2/AMPK/p38/p-Hsp27 axis in cell lines that ectopically express SDHA and B?

Response: We agree with the reviewer that an increase in p38 kinase is not limited to ARPI response but modulates different cellular processes. Our previous studies have shown that androgen increases Hsp27 phosphorylation through p38 (Zoubeidi *et al*, 2007) and enzalutamide

also increases p-Hsp27 levels in LNCaP and MR49F cells (Nappi *et al.*, 2020). We have incorporated a new panel in figure EV3 demonstrating that 12 hr of enzalutamide increases p-Hsp27 and p-p38 levels in LNCaP cells (please see the figure in the response to point 5 above). In this article, we have further focused on AR reactivation by energetic stress response via Hsp27 as an AR co-chaperone. Silencing of SDHA and SDHB subunits increased Ca²⁺/p-CAMKK2/p-AMPK/p-p38/p-Hsp27 axis in LNCaP cells. Overexpression of SDHA and SDHB subunits in LNCaP cells, as shown in figures 4D and 4E lower panels, downregulates p-CAMKK2 and p-AMPK levels. Similarly, Figure EV3E shows that overexpression of SDHB subunit decreases p-P38 and p-Hsp27 levels. SDHB silencing is known to increase p38 (Cervera *et al.*, 2008; Chen *et al.*, 2014) and AMPK phosphorylation (Chen *et al.*, 2014) and its overexpression in colorectal HT-29 cells inhibits AMPK phosphorylation (Xiao *et al.*, 2018).

7. It is unclear why SDHA kd affects SDHB levels (fig 3e). Authors may consider using 2-3 different siRNAs.

Response: SDHA and SDHB are subunits of a complex and silencing of one subunit affects the expression of other subunits as well (Lemarie *et al.*, 2011). Saxena *et al.* reported that genomic mutation in the *SDHB* gene in kidney cancer cell line UOK269 not only affected SDHB expression but also SDHA levels, and reintroducing *SDHB* gene also restored SDHA levels in UOK269WT (Saxena *et al.*, 2016). Therefore, reduced levels of SDHB after SDHA knockdown are not an off-target effect.

8. It is shown that ARPI-induced repression of SDH activity leads to adaptive increases in AR protein levels, it is unclear as to why this increase is only observed at earlier time points following ENZA (fig 2B) and not at later time points.

Response: As an acute adaptive response, the effect on AR protein level is detectable shortly after ARPI treatment. Unliganded AR is destabilized and therefore starts undergoing degradation. Being directly regulated through AREs, expression of SDH subunits and therefore SDH activity of the complex is affected leading to succinate accumulation as early as 12 hr, which eventually plays a protective role by stabilizing the AR protein. Hence, we see AR levels affected only at earlier time points following ENZA and not at later time points.

9. Authors suggest that "castration-resistant V16D and ENZA-resistant MR49F cell lines which maintain AR activity independent of androgen availability demonstrate more SDH activity compared to castration sensitive LNCaP cells". It is unclear what happens to the SDH-AR axis in these resistant cell lines? Would increased SDH activity not be sufficient to clear succinate and eventually block AR activation- is this not counterintuitive?

Response: AR restoration through succinate is an early cytoprotective approach to stabilize major driver and oncogene in AR-dependent prostate cancer cells, however these AR genomically altered V16D and ENZA-resistant MR49F cells (e.g. AR amplification, point mutations, or splice variants lacking ligand binding domains (Wyatt & Gleave, 2015) are pre-adapted to be resistant to AR

antagonists targeting the ligand binding domain, which enable them to grow even in the absence of ligand.

AR levels are high in these AR^{mut} resistant cells (and not reduced by AR antagonism), which increases SDH levels via its AREs, which should be enough to clear succinate levels. However, the active AR signaling in these cells increases p-CAMKK2 (Massie *et al*, 2011) and therefore drives activation of downstream p-AMPK/p-p38/p-Hsp27 axis as seen in human TMA and LNCaP xenograft studies. A recent study by Ware et al also demonstrates that enzalutamide-resistant cells derived from LNCaP, CS2, and LN95 cell lines increase p38 phosphorylation as an adaptive response to stress (<https://doi.org/10.1101/2020.04.22.050385>). Stress-activation of p-Hsp27 helps maintain AR levels even in the absence of accumulated succinate.

10. Data show "SDH-repressed cells demonstrated higher bioenergetic profiles by primarily increasing glycolysis (up to 100% increase)" how would they explain increased SDH activity observed in CRPC cell lines (Supp fig 3e). Does that mean CRPC is less reliant on glycolysis?

Response: CRPC cells show robust metabolic remodeling supported by various pathways including glycolysis, mitochondrial respiration, lipid, and amino acid metabolism to fulfil their high energy and macromolecule requirements. Metabolic profiling by Kaushik et al showed glycolysis, pyruvate metabolism, TCA cycle, pentose phosphate pathway, starch/sucrose, amino acid, acetate, and steroid metabolism as metabolic signatures of CRPC (Kaushik *et al*, 2014). CRPC cells are more 18F-fluorodeoxyglucose (18F-FDG) PET-positive compared to treatment naïve prostate cancer suggesting that they are more glycolytic. PI3K/Akt signaling activated in CRPC results in activation of Hexokinase (Martin *et al*, 2017) and lipogenic enzymes (Zadra *et al*, 2013) promoting glycolysis and lipid metabolism in CRPC. CRPC cells are more dependent on amino acids including Leucine (Tee, 2013), glutamine, alanine (Kaushik *et al.*, 2014; Zadra *et al.*, 2013) to feed the TCA cycle to generate energy and citrate for lipid metabolism. Once reactivated in CRPC, AR would fuel these pathways either through direct transcriptional regulation (Massie *et al.*, 2011), AMPK-PGC1alpha axis (Tennakoon *et al*, 2014) or metabolism enzymes/transcription factors (like FAS and SREBP) (Heemers *et al*, 2006; Swinnen *et al*, 1997). Our GSEA analysis also clearly showed enrichment of oxidative phosphorylation, adipogenesis, and fatty acid metabolism in AR-positive CRPCs compared to AR-negative NEPC patients (Figure EV 5E-F).

11. Inconsistent: fig 3e shows p-Hsp7 levels are enhanced following SDHA or B kd; however, in figure 4c SDHA/SDHB kd doesn't seem to decrease p-Hsp27 levels, please clarify.

Response: Similar to figure 3e, silencing of SDHA and SDHB subunits in figure 4C (lane 2 and lane 6) show increased p-Hsp27 levels compared to siScr (lane 1 and lane 5, respectively) under ENZA.

Fig. 3E

Fig. 4C

13. Authors show (supp fig 5B) lower AR levels/pHsp27 levels in NEPC, it would be interesting to test whether this is due to upregulation of SDH activation independent of AR axis in such tumors. Authors should also explain increased SDHA activity observed in CRPC patients (fig 5c).

Response: NEPC tumours are highly proliferative and glycolytic non-adenocarcinomas that have shed their dependence on canonical AR-driven pathways. As shown in suppl. Fig. 5C and D, NEPC PDXs have lower SDH levels compared to castrate-sensitive and -resistant adenocarcinomas. Our GSEA analysis also clearly showed that SDHB-related pathways oxidative phosphorylation, adipogenesis, and fatty acid metabolism were enriched only in AR-positive, castrate resistant adenocarcinomas but not AR-negative NEPC (Fig EV 5F). Reduced SDH activity usually leads to increased hypoxia and glycolysis (Cervera *et al.*, 2008; Saxena *et al.*, 2016), which are enriched in NEPC tumours (Fig. EV 5E). While causes for reduced AR levels in NEPC are multi-faceted (e.g. genomic, epigenomic, translation, post-translational), our data suggests that lower levels of the AR co-chaperone p-Hsp27, despite lower SDH levels, would lead to lower AR protein stability.

Succinate signaling helps adapt prostate cancer cells acutely to ARPI stress by stabilizing AR protein levels. Once adapted to the acute stress, clonal populations of prostate cancer cells can reactivate AR via genomic (amplification, variants, mutations) and metabolic (intra-tumoral steroidogenesis (Wyatt & Gleave, 2015) adaptations. The increased activity of SDH in CRPC reflects the reactivation of AR by these genomic or metabolic adaptations, that in turn transactivate SDH through AREs resulting in increased SDH activity. Additionally, AMPK-PGC1alpha axis activated by AR signaling may also lead to increased SDH activity in CRPCs (Tennakoon *et al.*, 2014).

REFREE 3:

The review of the article shows that the manuscript meets all required qualities for Embo Mol Med publication level. Technical, novelty and medical aspect are strong enough to go ahead with this study.

Comments for the Authors:

Here, Saxena et al. report the crucial role of succinate dehydrogenase (SDH) activity in reactivating AR signaling following an Androgen Receptor (AR) pathway inhibition (ARPI) strategy. Authors show that TCA cycle, and especially SDH A and B subunits, are controlled by AR through a "classical" transcriptional regulation. Reciprocally, authors show that inhibition of SDH results in AR protein accumulation and activity underlying a potential role in ARPI adaptation. Using mechanistic approaches, the authors demonstrated that succinate accumulation, resulting from SDH inhibition induced by ARPI, activated a calcium-dependent axis sustained by CAMKK2, AMPK, p38-MAPK and HSP27 to promote AR stabilization and to bypass AR inhibition. These findings have been correlated with human TMA investigations. Finally, the downstream target HSP27 inhibition, using the small inhibitor Ivermectin, has been challenged using a preclinical xenograft model. The experiments are well designed and performed; the statistical analysis meets sufficiency requirements. The manuscript is well written, and the results are extremely relevant as they proposed an early metabolic adaptation of PCa tumor cell to Androgen Deprivation Therapy. Biological questions are rigorously addressed using a wide range of experimental approaches that strengthened research rational and experimental workflow.

Although the manuscript is consistent and well conducted, some issues should be addressed to make the study more robust and more attractive for non-specialist readers.

Response: We thank the reviewer for the positive comments on our manuscript and its suitability for publication in the *EMBO Molecular Medicine*.

Major Issues

1) Paper rational is based on the central role of SDHA/B in acute ARPI resistance mechanism. Although the authors provide really clear-cut in vivo experiments using Ivermectin to target HSP27, pre-clinical investigation missed in vivo targeting of SDHA/B to validate succinate level as a critical crossroad to manage ARPI resistance. Thus, authors should test xenograft models using LNCaP cells or another cell line that overexpress SDHA/B and challenge Enzalutamide effect.

Response: We appreciate this suggestion but instead used untreated and post-ARPI treated human prostate cancers spotted on to TMA to help credentialize this pathway, which we believe is more relevant than using another xenograft model.

2) Extrapolation of data regarding SDHA/B status and Androgen Regulated Genes (ARG) signature should be investigated using public dataset in order to correlate SDHA/B expression with ARG. GSEA analysis should be an option.

Response: We performed GSEA analysis on AR-positive CRPC vs AR-negative NEPC and incorporated these results in figure EV 5 E-F. The analysis demonstrates enrichment of signature pathways including androgen response in CRPCs and G2/M checkpoint and E2F signaling in NEPC, respectively (Fig. EV 5E). Additionally, SDHB-related pathways including oxidative phosphorylation, adipogenesis and fatty acid metabolism were enriched in CRPC but not NEPC (Fig. EV 5F). Conversely, NEPC samples showed enrichment of hypoxia and glycolysis (Fig. EV 5E) which are known to be upregulated on SDH repression (Cervera *et al.*, 2008; Saxena *et al.*, 2016). This analysis from GSEA supports our experimental data and postulates.

3) Cell culture model limitations come from population clonality and cell specific response. Thus, authors should determine SDH activity and succinate accumulation (as depicted in Figure 1c/d) in others AR-positive prostate cancer cell lines as these experiments have been performed only in LNCaP cells. Authors should include at least similar experiments using LAPC4 cell line.

Response: We agree with the reviewer regarding limitation of the in vitro model in representing the diversity of human tumours. In addition to LNCaP, we included LAPC4 cells to evaluate the effect of ARPI treatment on SDH subunits in Fig.1B and effect of SDHA/B silencing (Fig. 2A) and overexpression (Fig. EV 2H) on AR levels.

In response to the reviewer's request, we added determination of succinate activity and intracellular succinate in LAPC4 cells, showing similar results to LNCaP cells that ARPI decreases SDH activity while increasing intracellular succinate (Fig 1C, 1D).

Minor Issues:

1) Figure 1F: Authors should provide statistical analysis for the ChIP assay. Y-axis legend should be corrected and mentioned "relative enrichment analysis/quantification" instead of "Relative mRNA levels"

Response: Statistical analysis has been added to the ChIP assay and Y-axis legend has been corrected to "relative enrichment analysis/quantification".

2) Figure S1a: SDHA results obtained with Mass Spectrometry approach exhibit an increased accumulation after 24 hours Enzalutamide exposition. Do the authors have an explanation for this result?

Response: To clarify the proteomics data illustrated in Fig. S1a shows that SDHA levels decrease after 24 h of ENZA treatment. Because AR directly regulates expression of SDHA and SDHB

subunits through AREs, acute treatment with ENZA for 24 affects mRNA and protein levels of SDH subunits and therefore SDH activity resulting in accumulation of succinate.

3) ARE binding site is characterized by an IR3 binding-site; hexamers half-site should be indicated by uppercase letters in Fig S1F.

Response: IR3 Hexamer half-sites have been indicated by uppercase letters in Fig S1F (mentioned as EV 1F in the revised manuscript).

4) Figure 5a,b,c: The significance NHT, abbreviation corresponding to "neoadjuvant-treated patients", needs to be indicated in the text legend.

Response: NHT abbreviation has been defined in the text legend.

5) Figure 3c, Figure 6c,d : Some of real time cell proliferation plots miss the control curve. Homogenization of the curve color charts between each plot should facilitate analysis. Statistical analysis should be included at least for the final time point.

Response: Control curves have been incorporated in Figure 3C, and Fig. 6C-D and statistical analysis has been indicated for the final time point.

6) A recent paper pointed out the key role of succinate accumulation in high-grade prostate cancer samples. This study provides important complementary results that strengthen the present article message. This article should be added in the manuscript and discussion. Nat Commun. 2020 Mar 20;11(1):1487.doi: 10.1038/s41467-020-15237-5. PMID: 32198407.

Response: This article has been referenced and described in the introduction and discussion section.

References:

- Aguiar CJ, Andrade VL, Gomes ER, Alves MN, Ladeira MS, Pinheiro AC, Gomes DA, Almeida AP, Goes AM, Resende RR *et al* (2010) Succinate modulates Ca(2+) transient and cardiomyocyte viability through PKA-dependent pathway. *Cell Calcium* 47: 37-46
- Ajdukovic J (2016) HIF-1--a big chapter in the cancer tale. *Exp Oncol* 38: 9-12
- Cervera AM, Apostolova N, Crespo FL, Mata M, McCreath KJ (2008) Cells silenced for SDHB expression display characteristic features of the tumor phenotype. *Cancer Res* 68: 4058-4067
- Chen L, Liu T, Zhang S, Zhou J, Wang Y, Di W (2014) Succinate dehydrogenase subunit B inhibits the AMPK-HIF-1alpha pathway in human ovarian cancer in vitro. *J Ovarian Res* 7: 115
- Coleman DJ, Van Hook K, King CJ, Schwartzman J, Lisac R, Urrutia J, Sehrawat A, Woodward J, Wang NJ, Gulati R *et al* (2016) Cellular androgen content influences enzalutamide agonism of F877L mutant androgen receptor. *Oncotarget* 7: 40690-40703

Glick D, Barth S, Macleod KF (2010) Autophagy: cellular and molecular mechanisms. *J Pathol* 221: 3-12

Heemers HV, Verhoeven G, Swinnen JV (2006) Androgen activation of the sterol regulatory element-binding protein pathway: Current insights. *Mol Endocrinol* 20: 2265-2277

Kaushik AK, Vareed SK, Basu S, Putluri V, Putluri N, Panzitt K, Brennan CA, Chinnaiyan AM, Vergara IA, Erho N *et al* (2014) Metabolomic profiling identifies biochemical pathways associated with castration-resistant prostate cancer. *J Proteome Res* 13: 1088-1100

Kumar A, Davuluri G, Welch N, Kim A, Gangadhariah M, Allawy A, Priyadarshini A, McMullen MR, Sandlers Y, Willard B *et al* (2019) Oxidative stress mediates ethanol-induced skeletal muscle mitochondrial dysfunction and dysregulated protein synthesis and autophagy. *Free Radic Biol Med* 145: 284-299

Lemarie A, Huc L, Pazarentzos E, Mahul-Mellier AL, Grimm S (2011) Specific disintegration of complex II succinate:ubiquinone oxidoreductase links pH changes to oxidative stress for apoptosis induction. *Cell Death Differ* 18: 338-349

Martin PL, Yin JJ, Seng V, Casey O, Corey E, Morrissey C, Simpson RM, Kelly K (2017) Androgen deprivation leads to increased carbohydrate metabolism and hexokinase 2-mediated survival in Pten/Tp53-deficient prostate cancer. *Oncogene* 36: 525-533

Massie CE, Lynch A, Ramos-Montoya A, Boren J, Stark R, Fazli L, Warren A, Scott H, Madhu B, Sharma N *et al* (2011) The androgen receptor fuels prostate cancer by regulating central metabolism and biosynthesis. *EMBO J* 30: 2719-2733

Mills E, O'Neill LA (2014) Succinate: a metabolic signal in inflammation. *Trends Cell Biol* 24: 313-320

Nappi L, Aguda AH, Nakouzi NA, Lelj-Garolla B, Beraldi E, Lallous N, Thi M, Moore S, Fazli L, Battsogt D *et al* (2020) Ivermectin inhibits HSP27 and potentiates efficacy of oncogene targeting in tumor models. *J Clin Invest* 130: 699-714

Saxena N, Maio N, Crooks DR, Ricketts CJ, Yang Y, Wei MH, Fan TW, Lane AN, Sourbier C, Singh A *et al* (2016) SDHB-Deficient Cancers: The Role of Mutations That Impair Iron Sulfur Cluster Delivery. *J Natl Cancer Inst* 108

Swinnen JV, Esquenet M, Goossens K, Heyns W, Verhoeven G (1997) Androgens stimulate fatty acid synthase in the human prostate cancer cell line LNCaP. *Cancer Res* 57: 1086-1090

Tee AR (2013) Metastatic castration-resistant prostate cancer hungers for leucine. *J Natl Cancer Inst* 105: 1427-1428

Tennakoon JB, Shi Y, Han JJ, Tsouko E, White MA, Burns AR, Zhang A, Xia X, Ilkayeva OR, Xin L *et al* (2014) Androgens regulate prostate cancer cell growth via an AMPK-PGC-1 α -mediated metabolic switch. *Oncogene* 33: 5251-5261

Wadosky KM, Koochekpour S (2017) Androgen receptor splice variants and prostate cancer: From bench to bedside. *Oncotarget* 8: 18550-18576

Wyatt AW, Gleave ME (2015) Targeting the adaptive molecular landscape of castration-resistant prostate cancer. *EMBO Mol Med* 7: 878-894

Xiao Z, Liu S, Ai F, Chen X, Li X, Liu R, Ren W, Zhang X, Shu P, Zhang D (2018) SDHB downregulation facilitates the proliferation and invasion of colorectal cancer through AMPK functions excluding those involved in the modulation of aerobic glycolysis. *Exp Ther Med* 15: 864-872

Yu K, Xiang L, Li S, Wang S, Chen C, Mu H (2019) HIF1 α promotes prostate cancer progression by increasing ATG5 expression. *Anim Cells Syst (Seoul)* 23: 326-334

Zadra G, Photopoulos C, Loda M (2013) The fat side of prostate cancer. *Biochim Biophys Acta* 1831: 1518-1532

Zhang F, Kumano M, Beraldi E, Fazli L, Du C, Moore S, Sorensen P, Zoubeidi A, Gleave ME (2014) Clusterin facilitates stress-induced lipidation of LC3 and autophagosome biogenesis to enhance cancer cell survival. *Nat Commun* 5: 5775

Zhou B, Kreuzer J, Kumsta C, Wu L, Kamber KJ, Cedillo L, Zhang Y, Li S, Kacergis MC, Webster CM *et al* (2019) Mitochondrial Permeability Uncouples Elevated Autophagy and Lifespan Extension. *Cell* 177: 299-314 e216

Zoubeidi A, Zardan A, Beraldi E, Fazli L, Sowery R, Rennie P, Nelson C, Gleave M (2007) Cooperative interactions between androgen receptor (AR) and heat-shock protein 27 facilitate AR transcriptional activity. *Cancer Res* 67: 10455-10465

We hope that you will now find our manuscript suitable for publication in *EMBO Molecular Medicine*, and we look forward to your reply.

25th Jan 2021

Dear Dr. Gleave,

Thank you for the submission of your revised manuscript to EMBO Molecular Medicine. We have now received the enclosed reports from the two referees who re-reviewed your manuscript. As you will see, they are both supportive of publication, and I am therefore pleased to inform you that we will be able to accept your manuscript once the following editorial points will be addressed:

1) Main manuscript text:

- Please answer/correct the changes suggested by our data editors in the main manuscript file (in track changes mode). This file will be sent to you in the next few days. Please use this file for any further modification.
- Please remove the coloured text.
- References: please adjust format so as to have 10 authors listed before et al.
- Material and methods:
 - o Please indicate the sequence/reference for the control siRNA used in your experiments (siScr).
 - o Thank you for providing antibodies information, please also make sure that the dilutions used in your study are indicated for each antibody.
 - o Please add a paragraph on Human subjects, and include a statement that informed consent was obtained from all subjects and that the experiments conformed to the principles set out in the WMA Declaration of Helsinki and the Department of Health and Human Services Belmont Report. Please also identify the committee approving the study protocol.
- Statistics: Please indicate in the figures or in the legends the exact n= and exact p= values, not a range, along with the statistical test used. Some people found that to keep the figures clear, providing a supplemental table (in the Appendix) with all exact p-values was preferable. You are welcome to do this if you want to.
- Please include a Data Availability Section:

Before submitting your revision, primary datasets produced in this study data need to be deposited in an appropriate public database (see <https://www.embopress.org/page/journal/17574684/authorguide#dataavailability>). Please note that the Data Availability Section is restricted to new primary data that are part of this study. If not applicable, please indicate in this section: "This study includes no data deposited in external repositories"

2) Figures:

As mentioned in a previous correspondence, please update Figure 4 as indicated (using a similar +/- system as in the source data instead of flipping and reusing the bands, or disposing the lanes in the desired order, without reusing the control, and clearly indicating the cuts in the figure. Another alternative would be to present another blot or re-run the blots with the samples loaded in a more logical order.) Please also update the associated Source Data file. Please also indicate whether SiScr has been used as control in Figure 4C-G.

3) As part of the EMBO Publications transparent editorial process initiative (see our Editorial at <http://embomolmed.embopress.org/content/2/9/329>), EMBO Molecular Medicine will publish online a Review Process File (RPF) to accompany accepted manuscripts.

In the event of acceptance, this file will be published in conjunction with your paper and will include

the anonymous referee reports, your point-by-point response and all pertinent correspondence relating to the manuscript. Let us know whether you agree with the publication of the RPF and as here, or IF YOU WANT TO REMOVE OR NOT ANY FIGURES FROM IT prior to publication. Please note that the Authors checklist will be published at the end of the RPF.

I look forward to receiving your revised manuscript.

Yours sincerely,

Lise Roth

Lise Roth, PhD
Editor
EMBO Molecular Medicine

***** Reviewer's comments *****

Referee #2 (Remarks for Author):

The authors have now revised the manuscript addressing all points raised and included new experiments to support their findings in response to the comments raised.

Referee #3 (Comments on Novelty/Model System for Author):

Quality of the manuscript has been greatly improve through revision process. The various models provide to conduct these experiments reach this article to a high level in the field of prostate cancer.

Referee #3 (Remarks for Author):

The answers provide in the rebuttal letter, scientific improvement of the manuscript and additional experiments performed, greatly increase manuscript quality. In my opinion, current version of the article is now suitable for publication in EMBO Molecular Medicine. We greatly appreciate the work that have been done for the revision of the manuscript and that authors taking care of advices provided.

The authors performed the requested editorial changes.

9th Feb 2021

Dear Dr. Gleave,

Thank you for submitting your revised manuscript. I looked at everything and all is fine. I am thus very pleased to accept your manuscript for publication in EMBO Molecular Medicine!

It will now be sent to our publisher to be included in the next available issue of EMBO Molecular Medicine.

Please read below for additional important information regarding your article, its publication and the production process.

Congratulations on an interesting study!

Yours sincerely,

Lise Roth

Lise Roth, Ph.D
Editor
EMBO Molecular Medicine

Follow us on Twitter @EmboMolMed
Sign up for eTOCs at embopress.org/alertsfeeds

Corresponding Author Name: Martin Gleave

Manuscript Number: EMM-2020-13427-V2